# The combination of genomic offset and niche modelling provides insights into climate change-driven vulnerability

Yilin Chen [1,2,6], Zhiyong Jiang [1,2,6], Ping Fan [1,2], Per G. P. Ericson [3], Gang Song[1], Xu Luo[4], Fumin Lei [1,2,5] ✉ & Yanhua Qu [1,2] ✉

Global warming is increasingly exacerbating biodiversity loss. Populations locally adapted to spatially heterogeneous environments may respond differentially to climate change, but this intraspecific variation has only recently been considered when modelling vulnerability under climate change. Here, we incorporate intraspecific variation in genomic offset and ecological niche modelling to estimate climate change-driven vulnerability in two bird species in the Sino-Himalayan Mountains. We found that the cold-tolerant populations show higher genomic offset but risk less challenge for niche suitability decline under future climate than the warm-tolerant populations. Based on a genome-niche index estimated by combining genomic offset and niche suitability change, we identified the populations with the least genome-niche interruption as potential donors for evolutionary rescue, i.e., the populations tolerant to climate change. We evaluated potential rescue routes via a landscape genetic analysis. Overall, we demonstrate that the integration of genomic offset, niche suitability modelling, and landscape connectivity can improve climate change-driven vulnerability assessments and facilitate effective conservation management.

Anthropogenic climate change is one of the primary drivers of environmental change and global biodiversity loss[1,2]. Rapid climate changes can lead to shifts in species ranges, population decline and even extinction if the changes exceed the physiological tolerance of organisms[3,4]. As these effects become increasingly profound in light of global warming, how organisms can respond to environmental changes is becoming a focus of basic research[5,6]. Modelling changes in the distribution range, suitable climatic conditions and vegetation types of species under different climate scenarios has provided considerable insights into the impacts of climate change on biodiversity (e.g. refs. 6–10). However, as these models rely solely on abiotic and biotic environmental changes, ecologic genomics, which investigates how the genomic variants of a species vary along current environmental gradients and how much genetic change has been required to keep up with climate change-driven environmental changes, has only recently been integrated into modelling species responses to climate change[11–13].

These ecological genomic studies have shown different genotype-climate associations between populations and suggested that local populations may respond differentially to climate change[11–13]. Nevertheless, intraspecific variation has rarely been incorporated into ecological niche models that often assume a uniform climate response between populations[11,14–16]. There is an urgent need to incorporate intraspecific genomic variation in modelling habitat suitability in the

[1]Key Laboratory of Zoological Systematics and Evolution, Institute of Zoology, Chinese Academy of Sciences, Beijing, China. [2]College of Life Sciences, University of Chinese Academy of Sciences, Beijing, China. [3]Department of Bioinformatics and Genetics, Swedish Museum of Natural History, PO Box 50007, SE-104 05 Stockholm, Sweden. [4]Faculty of Biodiversity and Conservation, Southwest Forestry University, Kunming, China. [5]Center for Excellence in Animal Evolution and Genetics, Chinese Academy of Sciences, Kunming 650223, China. [6]These authors contributed equally: Yilin Chen, Zhiyong Jiang. ✉e-mail: leifm@ioz.ac.cn; quyh@ioz.ac.cn

context of climate change, as such information is necessary for understanding fine-scale estimates of climate change-driven vulnerability (e.g. ref. 17).

Mountainous areas harbour exceptional biodiversity and endemism but are highly vulnerable to climate change[18]. This is because the complex topography within a rather small geographical area leads to dramatic ecological stratification in mountain regions. Species are often confined to spatially heterogeneous environments and locally adapted to diversified climate conditions[18–20]. These population-specific adaptations likely drive different responses under climate change because the populations likely track their own optimal environmental conditions[20]. Despite having high potential, this intraspecific variation has not been considered in vulnerability estimates driven by climate change in mountainous species. Herein, we integrate ecological genomics and niche modelling to investigate the population-specific responses to future climate change in two mountainous birds in the Sino-Himalayan Mountains. Together with the mountains of southwestern China (hereafter the Southwest Mountains) and the East Himalayan hotspots[21], these areas harbour unique temperate biodiversity of high elevation but are now under serious threat from the rapidly changing climate[22].

Our focal species, the Green-backed tit (*Parus monticolus*) and Elliot's laughingthrush (*Trochalopteron elliotii*), are two mountainous birds that are mainly found in the Sino-Himalayan Mountains. The two species are distributed across different elevation ranges; *P. monticolus* is a mid-elevational bird commonly found between 1500 and 2500 m above sea level (a.s.l.), whereas *T. elliotii* lives at higher elevations between 2000 and 4500 m a.s.l.[23]. Previous phylogeographic studies have revealed that these birds are often restricted to ecologically and topographically heterogeneous areas, suggesting that they have adapted locally to disparate ecological zones[24–26]. As such, these species offer an excellent study system to examine the intraspecific variation in genotype-climate associations and how this would influence climate change-driven vulnerability.

Here, we combine ecological genomics and niche modelling to evaluate the population-specific responses of the two species to future climate change. We consider both the genomic offset (i.e., a measure

of the mismatch in genotype-climate association between current and potential future climates[11]) and niche suitability change (i.e., a measure of the difference in niche suitability between current and future ecological niches) for potential evolutionary rescue; that is, only populations with minor genome-niche interruption can serve as a desired store of species' survival for climate change (i.e., being a donor). We then use a landscape genetic approach to evaluate potential rescue costs (a framework is demonstrated in Fig. 1). Overall, as genomic offset and niche suitability change show different population-specific vulnerability estimates under future climate change, we emphasise the importance and strength of combining ecological genomics, niche modelling and landscape connectivity to guide effective mitigation efforts on increasingly threatened biodiversity.

## Results

### Species distribution
The distribution ranges of *T. elliotii* and *P. monticolus* cover the Sino-Himalayan Mountains and the mid-elevational mountains in Central China (Fig. 2a). Across their ranges, the average elevation declines from west to east and from north to south. The temperature and precipitation decrease from southeast to northwest and from south to north. The current distributions of *T. elliotii* and *P. monticolus* partially overlap but differ in certain ecological zones and elevation ranges (Fig. 2b). They both inhabit in the southwest mountainous zone, the western mountainous plateau zone, and parts of the Loess Plateau zone and the east meadow zone. However, *T. elliotii* also occurs in the southern Tibetan zone, whereas *P. monticolus* also lives in the eastern Himalayan zone. Given the considerable environmental and climatic variation across their distribution ranges, the two species provide an excellent opportunity to study intraspecific responses to environmental change following future climate change.

**De novo genome assembly and annotation of *T. elliotii*.** We generated a reference genome for *T. elliotii* to facilitate mapping of the resequencing data. In total, 157 gigabases (Gb) data were sequenced for a *T. elliotii* individual. After cleaning and quality control of the raw data, all qualified data were assembled into a genome of 1.12 Gb with an N50 contig length of 95 Kb and N50 scaffold length of 2.702 Mb. Using a homologue-based approach, we annotated 17,585 protein-coding genes. Using Benchmarking Universal Single-Copy Orthologs (BUSCO, aves_odb9[27]) as the reference gene set, we estimated that the assembly contains 90% complete single-copy BUSCOs, 2% complete duplicated BUSCOs, and 4.5% fragmented BUSCOs. We used this genome for the subsequent ecological genomic analyses.

**Intraspecific variation in genotype-climate association.** We integrated population genomics and climatic data to identify climate-associated genomic variation. We generated genome-wide resequencing data from 55 and 58 individuals across the distribution ranges of *T. elliotii* and *P. monticolus*, respectively (Supplementary Data 1). It is noteworthy that the strategy to obtain deep-sequenced whole-genome datasets (at average coverage of 19.14× and 15.48× for *T. elliotii* and *P. monticolus*, respectively, Supplementary Data 2) allowed us to capture maximal genomic variation, in contrast to most previous studies that are based on reduced-representation genomic datasets (see ref. 13). Using a de novo genome of *T. elliotii* generated in this study and an assembly of the great tit (*Parus major*) genome[28], we identified 10.3 and 3.9 million single-nucleotide polymorphisms (SNPs) in *T. elliotii* and *P. monticolus*, respectively.

We used gradientForest[29], a machine-learning regression tree-based approach, to first identify the climatic variables that are most closely associated with the genetic variation in the two species and then transform multidimensional climatic variables into multi-dimensional genetic space (i.e., turnover in allele frequencies)[11]. We found that climatic variables related to seasonal changes in

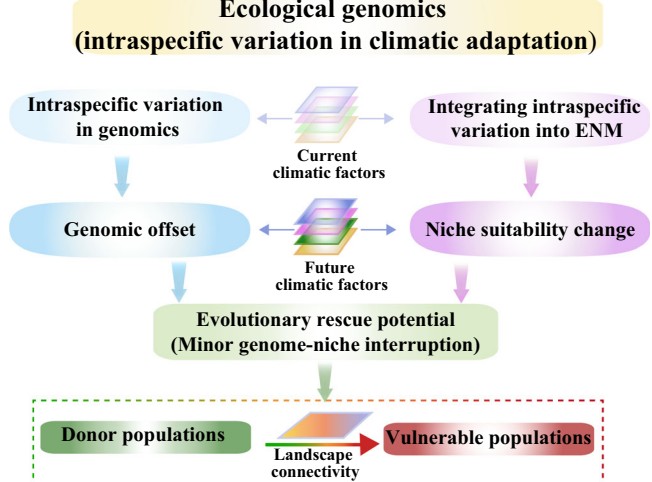

**Fig. 1 | A framework integrating ecological genomics, niche modelling and landscape genetic analysis to evaluate climate change-driven vulnerability.** A schematic representation of climate change-driven vulnerability modelling. We combine ecological genomics and niche modelling to evaluate the population-specific responses of the two species to future climate change. We consider both the genomic offset and niche suitability change for potential evolutionary rescue. Only populations with minor genome-niche interruption can be regarded as a desired store of species' survival for climate change (i.e., being a donor). A landscape genetic approach is then used to evaluate potential rescue costs.

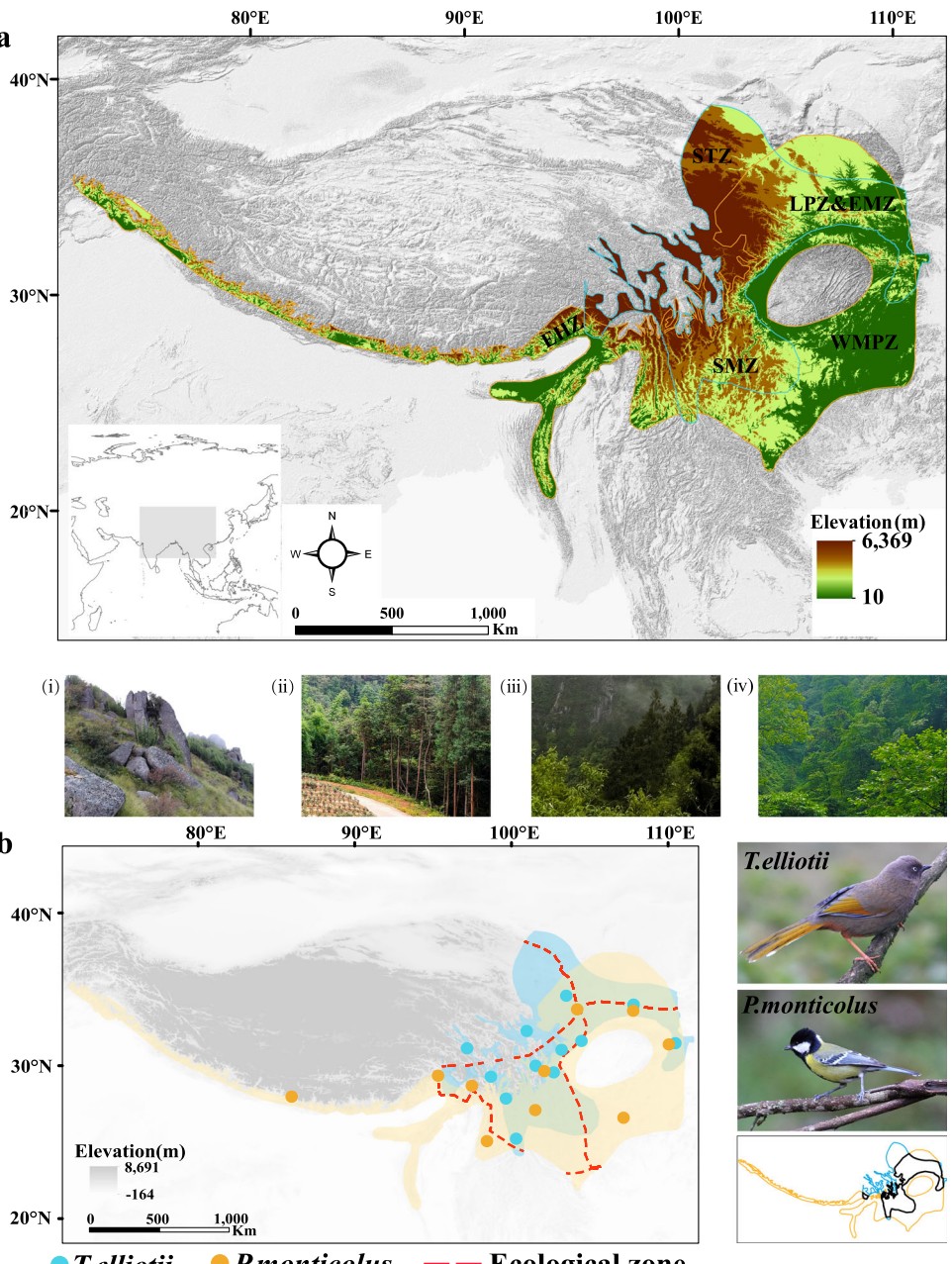

**Fig. 2 | Sino-Himalayan Mountains and distribution ranges of the two mountainous species studied. a** The distribution ranges of *P. monticolus* and *T. elliotii* cover the Sino-Himalayan Mountains and mid-elevational mountains in Central China. The southern Tibetan zone (STZ) is in the southeastern part of the Qinghai-Tibetan Plateau, which has an average elevation of 4500 m a.s.l. This zone is dominated by shrubland (i). The southwest mountainous zone (SMZ) has a vertical climatic zonation along an elevation gradient, ranging from the temperate coniferous forest (ii), mixed forests (iii) and subtropical broadleaf forest (iv). The eastern Himalayan zone (EHZ) is on the southern margin of the Qinghai-Tibetan Plateau and encompasses a broad range of ecological habitats varying from grassy meadows to a dense humid evergreen forest (i-iii). The western mountainous plateau zone (WMPZ) is on the east side of the Sino-Himalayan Mountains and is characterised by temperate broadleaf forests (iii). **b** Left, the distribution ranges of *P. monticolus* (yellow shade) and *T. elliotii* (blue shade) partially overlap in the southwest mountainous zone and western mountainous plateau zone, as well as in parts of the Loess Plateau zone and east meadow zone. In the non-overlapping parts of their distributions, *T. elliotii* occurs in the southern Tibetan zone, and *P. monticolus* lives in the eastern Himalayan zone. The yellow and blue dots show the sampling localities of *P. monticolus* and *T. elliotii* used in the genomic analysis. Right, upper, *T. elliotii*; middle, *P. monticulus*; bottom, distribution ranges of the *T. elliotii* and *P. monticulus* are shown by the blue and yellow outlines, respectively, while the areas where they overlap are shown by the black outline.

temperature and precipitation were strongly associated with the genomic variants in both species. Of the 19 climatic variables tested (Supplementary Table 1), the top five uncorrelated variables identified were BIO3 (isothermality, i.e., mean diurnal range divided by the temperature annual range), BIO18 (precipitation of the warmest quarter), BIO9 (mean temperature of the driest quarter), BIO19 (precipitation of the coldest quarter) and BIO5 (max temperature of the warmest month) for *P. monticolus*, while the top five climatic variables for *T. elliotii* were BIO2 (mean diurnal range, i.e., the mean of the monthly differences between the maximum and minimum temperatures), BIO10 (mean temperature of the warmest quarter), BIO7 (temperature annual range), BIO19 and BIO4 (temperature seasonality) (Fig. 3a, b and Supplementary Table 2).

Out of 50,000 randomly extracted SNPs, we identified 5446 and 7294 SNPs that were significantly associated with the top climatic variables ($R^2 > 0$, see Methods) for *T. elliotii* and *P. monticolus*,

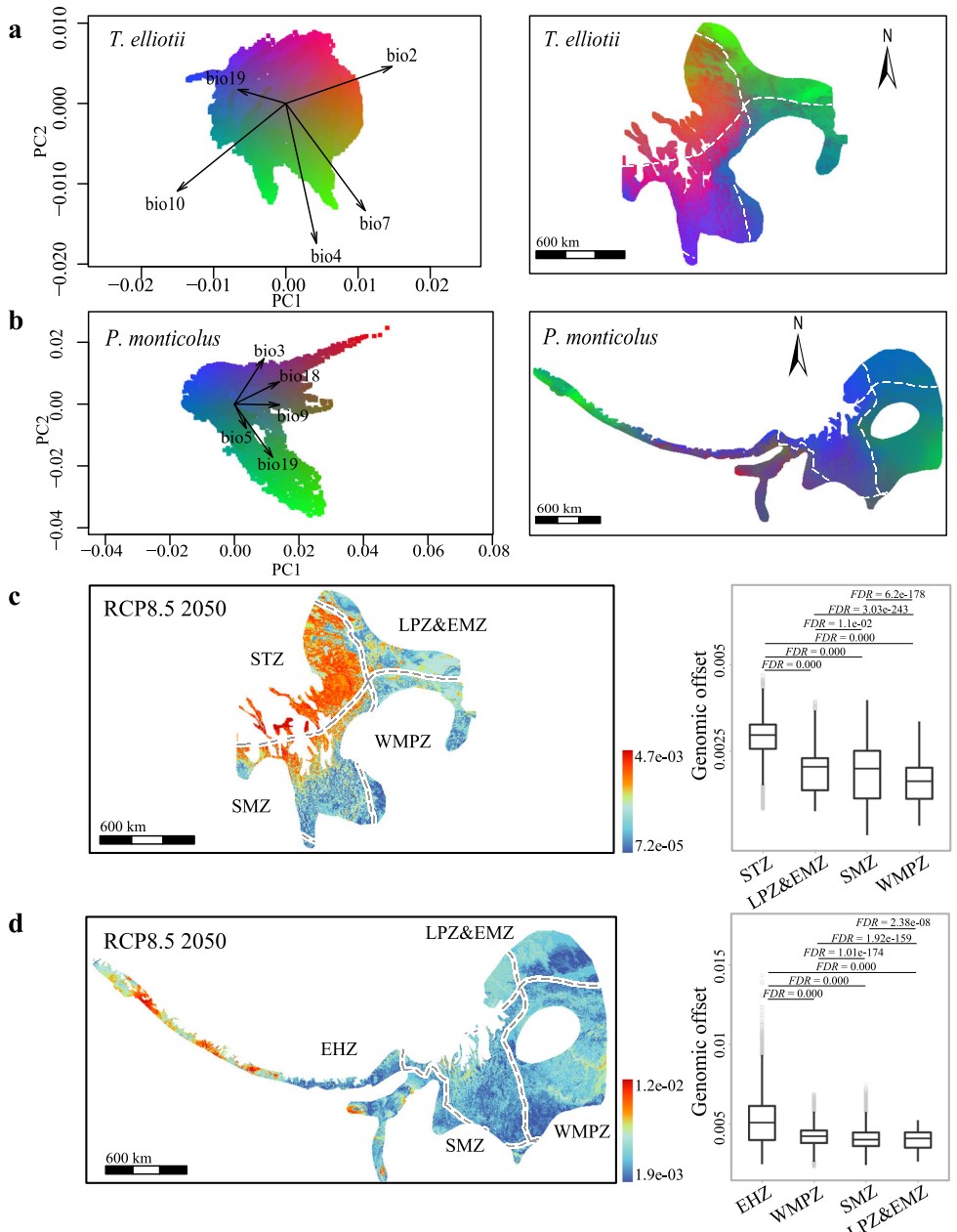

**Fig. 3 | GradientForest modelling genotype-climate associations and genomic offsets to future climate conditions of the two mountainous species. a**, **b** Left, principal component analyses of gradientForest transformed climatic variables for *T. elliotii* (**a**) and *P. monticolus* (**b**). Arrows show the loadings of the top-ranked uncorrelated climatic variables. The labelled vectors of the first two principal components indicate the direction and magnitude of correlation. Right, gradientForest transformed genotype-climate association across the distribution ranges of *T. elliotii* (**a**) and *P. monticolus* (**b**). The difference in genetic composition is mapped by assigning the three principal components to the RGB colour palette according to the gradientForest manual, with the resulting colour corresponding to the expected patterns of genetic composition. **c**, **d** Left, the gradientForest predicted genomic offsets under RCP8.5 2050 in *T. elliotii* (**c**) and *P. monticolus* (**d**). Right, populations in the western parts of the distributions, i.e., the southern Tibetan zone (STZ, *T. elliotii*, $n = 12,912$ grids) and eastern Himalayan zone (EHZ,

*P. monticolus*, $n = 9224$ grids), show greater genomic offsets than the populations inhabiting the eastern (LPZ&EMZ, Loess Plateau zone and east meadow zone, *T. elliotii*, $n = 8694$ grids, *P. monticolus*, $n = 10,551$ grids; WMPZ, western mountainous plateau zone, *T. elliotii*, $n = 6902$ grids, *P. monticolus*, $n = 26,979$ grids) and southern parts of the distribution ranges (SMZ, southwest mountainous zone, *T. elliotii*, $n = 14,981$ grids, *P. monticolus*, $n = 20,275$ grids). We tested genomic offsets among these groups using the two-tailed Wilcoxon rank-sum test and FDR correction for multiple comparisons. The box plots show the median (centre line) and 25th–75th percentiles (box limits). The whiskers extend to the top/bottom to the maxima and minima. Data beyond the end of the whiskers are considered outliers. White broken lines demonstrate ecological zones. See Supplementary Fig. 1 for the predicted genomic offsets under different emission scenarios and decades (RCP4.5 2050, RCP8.5 2050, RCP4.5 2070 and RCP8.5 2070).

respectively. Using an aggregate community-level turnover function across these SNPs, we visualised the genotype-climate turnover surface across the distribution ranges of the two species using the first three principal components (PCs) of the gradientForest outputs. We found that the genotype-climate associations vary from

the western parts (i.e., the southern Tibetan zone and eastern Himalayan zones) to the eastern (i.e., the western mountainous plateau zone) and southern parts of the distribution ranges (i.e., the southwest mountainous zone) (Fig. 3a, b). These results suggest that the two species show intraspecific variation in their genotype-

climate associations, which are likely subject to local adaptation to heterogeneous climatic conditions.

**Genomic offset to future climate change.** Based on the genotype-climate associations across the distribution ranges of the two species, we predicted which part of their ranges might be most vulnerable to future climate change. We used gradientForest[29] to calculate genomic offset as the Euclidean distance between current and projected future genomic compositions. Genomic offset is thus a measure of how much genetic change is needed to adjust to the new climate conditions, and the populations with the greatest genomic offset are those that have to adjust the most. To account for variability between climate models, we predicted the genomic offset for each species using four different climate models, MPI-ESM-LR[30], CCSM4[31], MICRO-5[32] and CNRM-CM5-2[33] (Supplementary Table 3), under the moderate scenario (representative concentration pathway, RCP 4.5 W/m²) and the worst scenario (RCP 8.5 W/m²) of greenhouse gas emission trajectories in 2050 and 5070, respectively. Genomic offset estimates revealed similar but magnitude-dependent spatial patterns under the four RCP emission scenarios considered at the 2050 and 2070 horizons (Supplementary Fig. 1). These results consistently showed that the populations in the western part (*T. elliotii* in the southern Tibetan zone and *P. monticolus* in the eastern Himalayan zone) exhibited greater genomic offset than the populations inhabiting the eastern (the Loess Plateau zone, eastern meadow zone and western mountainous plateau zone) and the southern parts of the distribution ranges (the southwest mountainous zone, two-tailed Wilcoxon rank-sum test, false discovery rate (FDR)-adjusted $P < 0.001$, Fig. 3c, d). Altogether, these results suggest that intraspecific variation in genotype-climate associations of the two species likely drives different genomic offsets in response to future climate change.

**Intraspecific variation in genotype-climate association drives different degrees of genomic offsets.** To investigate how the intraspecific variation in the genotype-climate association has driven the different degrees of genomic offsets in response to climate change, we identified genomic variants that are significantly associated with each of the top climatic variables (as identified by the gradientForest analysis) using three complementary approaches: latent factor mixed model (LFMM[34]), redundancy analysis (RDA) and distance-based redundancy analysis (dbRDA)[35,36]. A total of 72 and 798 SNPs were identified by all three methods for *T. elliotii* and *P. monticolus*, respectively (Fig. 4a, b). These SNPs are widely distributed across the genomes, with 25 and 204 SNPs located in the coding sequence and promoter regions (5k upstream and downstream of genes) across 23 and 147 genes for *T. elliotii* and *P. monticolus*, respectively (Supplementary Table 4). We then carried out functional enrichment analyses and found that these genes are functionally enriched in catalytic and metabolic processes (Supplementary Fig. 2), with some previously documented as being important in climate adaptation. In particular, *CRB1*, the only climate-associated gene identified in both *T. elliotii* and *P. monticolus*, has also been identified in other cold-tolerant vertebrates, i.e., the Adélie penguin (*Pygoscelis adeliae*)[37] and the Emperor penguin (*Aptenodytes forsteri*)[38]. The relationships between allele frequency variation in the SNP of *CRB1* and the mean diurnal range (BIO2, *T. elliotii*), as well as the isothermality (BIO3, *P. monticolus*), suggest a potential role of this gene in adaptation to extreme temperatures (Fig. 4c, d). In addition, we also detected some genes that are related to heat and cold tolerance, i.e., *VPS53*[39], *BRS3*[40] and *FASN*[41], as well as adaptation to hypoxic conditions, i.e., *MYH7*[42], *SLC19A1*[43] and *ARHGAP39*[44].

We then carried out population genetic structure analyses based on these SNPs using Admixture v1.3[45] to define the climate-tolerant groups. We found $K = 3$ to have the smallest cross-validation error in

both species and thus to be the optimal number of clusters explaining the variation among individuals (Supplementary Fig. 3). Based on the proportion of the respective cluster within each individual (>0.6), we classified 33% of *T. elliotii* individuals as tolerant to cold-dry conditions (shrubland and coniferous forests in the southern Tibetan zone and southwest mountainous zone), 15% as tolerant to warm-humid conditions (subtropical broadleaf forests in the southwest mountainous zone), and 16% to warm-dry conditions (temperate broadleaf forests in the western mountainous plateau zone). In *P. monticolus*, 10% of the individuals were classified as cold-dry tolerant (coniferous forests in an eastern Himalayan zone), 40% of the individuals were classified as warm-humid tolerant (subtropical broadleaf forests in the southwest mountainous zone), and 48% of the individuals were classified as warm-dry tolerant (temperate broadleaf forest in the western mountainous plateau zone, Fig. 4e, f).

We then compared genomic offset values among the three groups using outputs calculated from gradientForest analysis (i.e., $R^2$ positive SNP dataset, see Methods). We found that the cold-dry tolerant groups showed greater genomic offsets than the warm-humid and warm-dry tolerant groups (two-tailed Wilcoxon rank-sum test, FDR-adjusted $P < 0.001$, Fig. 4g, h and Supplementary Fig. 4a). In addition, we implemented a separate gradientForest analysis using the outlier SNPs identified by the three genotype-climate association analyses (i.e., outlier SNP dataset), and the analyses yielded similar results (two-tailed Wilcoxon rank-sum test, FDR-adjusted $P < 0.05$, Fig. 3i, j and Supplementary Fig. 4b). To further validate this, we also carried out a parallel genomic offset analysis using generalised dissimilarity modelling (GDM[46]), a distance-based method (i.e., $F_{ST}$) that is less sensitive to unequal sampling sizes, to estimate and compare genomic offsets among the three groups. Our GDM analyses using two datasets ($R^2$ positive SNP and outlier SNP datasets, see Methods) show similar patterns to those of the gradientForest analyses (two-tailed Wilcoxon rank-sum test, FDR-adjusted $P < 0.001$, Supplementary Fig. 5), suggesting that our genomic offset results are robust to different datasets and methods.

**Integrating intraspecific variation into ecological niche modelling.** We next investigated whether the predicted changes in niche suitability caused by future climate conditions differed among the three climate-tolerant groups. We carried out ecological niche modelling using an ensemble modelling approach that combines the outputs of different modelling algorithms, including maximum entropy, generalised boosted model, generalised additive model and multivariate adaptive regression splines implemented in Biomod2[47]. We projected niche suitability separately for the cold-dry, warm-humid and warm-dry tolerant groups, respectively. All ecological niche models have great discrimination ability (true skill statistics, TSS, 0.71–0.88; area under the receiver operating characteristic curve, AUC, 0.9–0.96, Supplementary Table 5).

Niche modelling inferred under the different future emission scenarios and decades consistently showed that the three climate-tolerant groups had different degrees of niche suitability change (Fig. 5a, b and Supplementary Fig. 6). Using the current niche suitability value as the benchmark, we calculated the change in the niche suitability index between the current and future climate (NSC = niche suitability index in the future climate − niche suitability index in the current climate). A negative value indicates a decrease, while a positive value shows an increase in niche suitability under future climate conditions. We found that the NSC decreased to a much smaller degree in those areas that harboured cold-dry tolerant groups than in those areas that maintained the warm-dry and warm-humid groups (two-tailed Wilcoxon rank-sum test, FDR-adjusted $P < 0.001$, Fig. 5a, b and Supplementary Fig. 6). These results suggest that the areas suitable for warm-tolerant individuals are more likely to decrease suitability under future

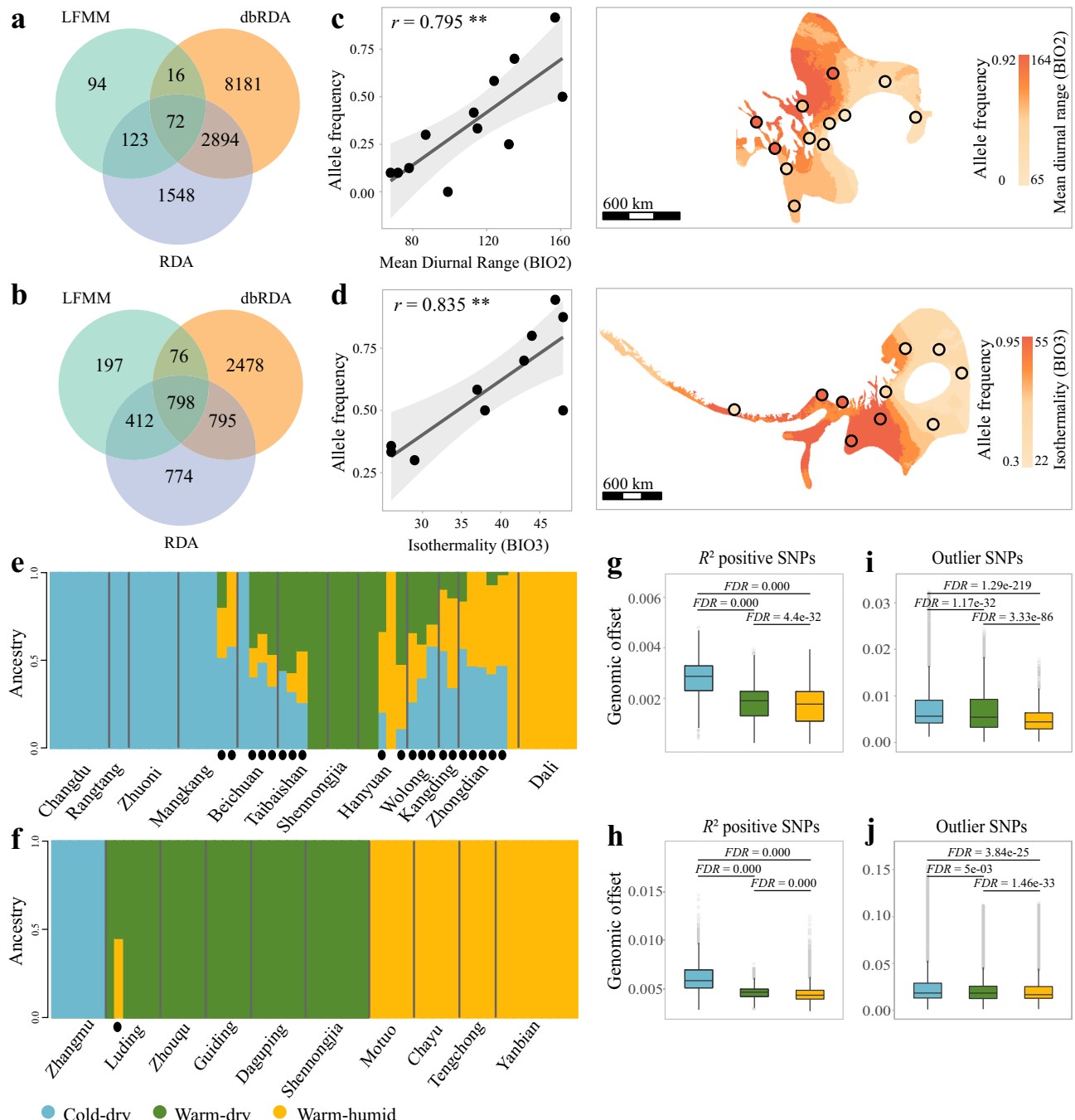

**Fig. 4 | Intraspecific variation in genotype-climate association. a, b** Venn diagram depicting overlaps between outlier single-nucleotide polymorphisms (SNPs) detected using LFMM, RAD and dbRAD in *T. elliotii* (**a**) and *P. monticolus* (**b**), respectively. **c, d** Relationships (left) and genotype-climate associations (right) between the allele frequency of SNP in *CRB1* and the mean diurnal range in *T. elliotii* (c, *n* = 55 individuals) and the isothermality in *P. monticolus* (d, *n* = 58 individuals). Light grey shade indicates the 95% confidence interval of the regression. **e, f** Population genetic structure plots of outlier SNPs separating groups of individuals tolerant to cold-dry (blue), warm-dry (green) and warm-humid conditions (yellow) in *T. elliotii* (**e**) and *P. monticolus* (**f**). The black dots show individuals with mixed genetic components. **g, j** The cold-dry tolerant groups (*T. elliotii*, *n* = 21,402 grids; *P. monticolus*, *n* = 6926 grids) show greater genomic offsets than the warm-humid (*T. elliotii*, *n* = 7061 grids; *P. monticolus*,

*n* = 30,336 grids) and warm-dry tolerant groups (*T. elliotii*, *n* = 15,416 grids; *P. monticolus*, *n* = 42,929 grids) in *T. elliotii* (**g, i**) and *P. monticolus* (**h, j**). We tested genomic offsets among these groups using the two-tailed Wilcoxon rank-sum test and FDR correction for multiple comparisons. **g, h** $R^2$ positive SNP datasets, **i, j** outlier SNP datasets. Note that spatial scales of the Y axis (genomic offsets) for the $R^2$ positive SNP datasets (**g, h**) differ from those for the outlier SNP datasets (**i, j**). **g–j** The box plots show the median (centre line) and 25th–75th percentiles (box limits). The whiskers extend from the top/bottom to the maxima and minima. Data beyond the end of the whiskers are considered outliers. Two-tailed Wilcoxon rank-sum test and FDR correction for multiple comparisons. Only the predicted genomic offsets under RCP8.5 2050 are shown, while those under the different emission scenarios and decades (i.e., RCP4.5 2050, RCP8.5 2050, RCP4.5 2070 and RCP8.5 2070) can be found in Supplementary Fig. 4.

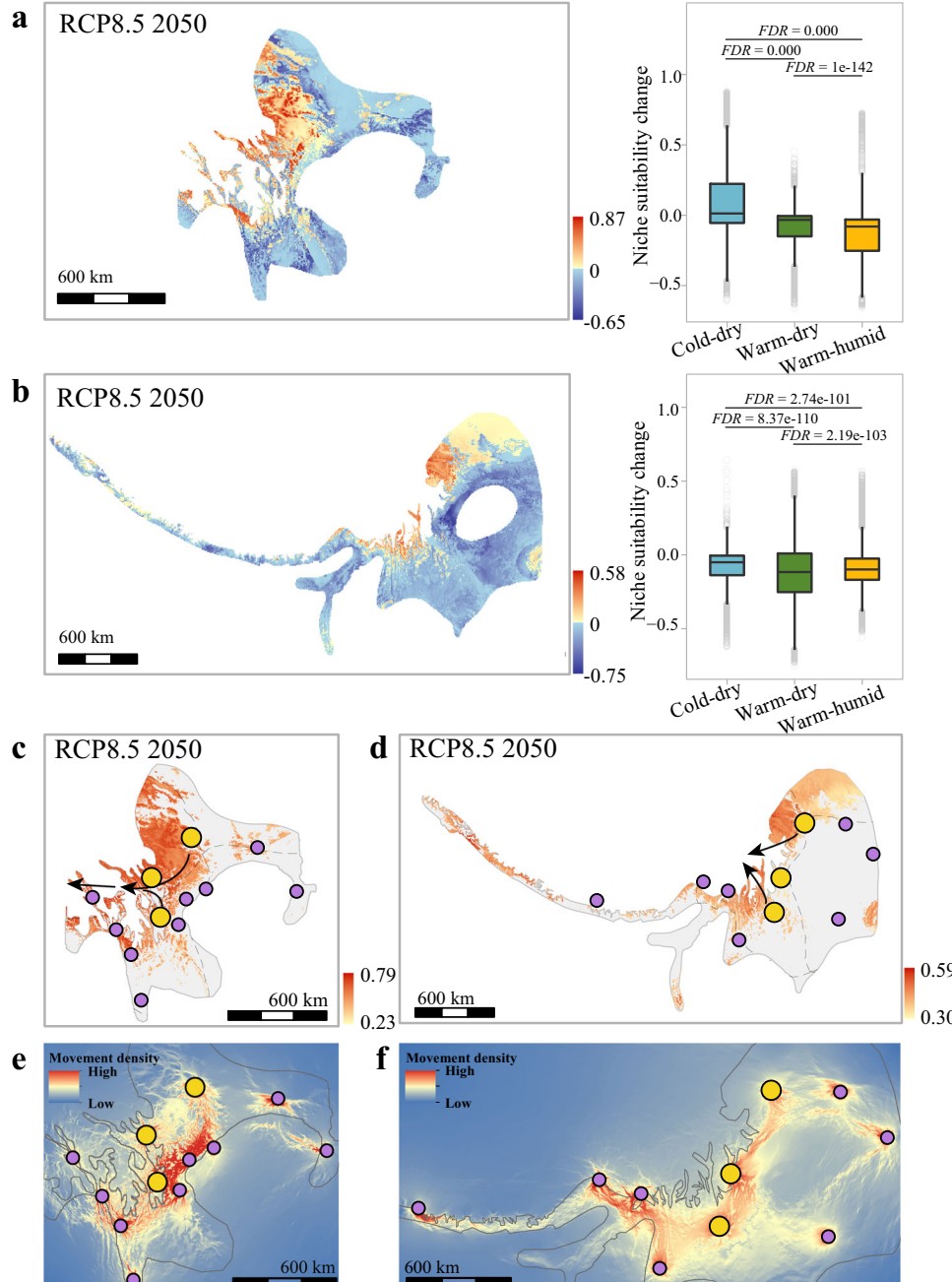

**Fig. 5 | Niche suitability change predicted by ecological niche modelling and evolutionary rescue routes identified by the landscape genetic analysis.**
**a, b** Left, projection of niche suitability change between current and future climate conditions under RCP8.5 2050 (niche suitability index predicted for future climate minus niche suitability index for current climate). The reddish colours show areas with increasing niche suitability (>0), and blue colours show areas with decreasing niche suitability (<0). Right, the warm-humid (*T. elliotii*, n = 7061 grids; *P. monticolus*, n = 30,336 grids) and warm-dry tolerant groups (*T. elliotii*, n = 15,416 grids; *P. monticolus*, n = 42,929 grids) are under the severer challenge for niche suitability decline than are the cold-dry tolerant groups (*T. elliotii*, n = 21,402 grids; *P. monticolus*, n = 6926 grids). We tested niche suitability change among these groups using the two-tailed Wilcoxon rank-sum test and FDR correction for multiple comparisons. The box plots show the median (centre line) and 25th–75th percentiles (box limits). The whiskers extend to the top/bottom to the maxima and

minima. **a** *T. elliotii*; **b** *P. monticolus*. **c, d** Projection of the genome-niche index based on the combined estimates of genomic offset and niche suitability change under RCP8.5 2050. The populations in the central areas of the Southwest Mountains have the least genome-niche interruption by climate change, and can then serve as the donor populations (large yellow dots) for evolutionary rescue. Black arrows show potential routes for evolutionary rescue under climate change.
**a–d** Only the projections under RCP8.5 2050 are shown, and those under different emission scenarios and decades (RCP4.5 2050, RCP8.5 2050, RCP4.5 2070 and RCP8.5 2070) can be found in Supplementary Figs. 6, 7. **e, f** Landscape genetic analysis predicted the density of dispersals between populations based on the effect of niche suitability for *T. elliotii* (**e**) and niche suitability and elevation for *P. monticolus* (**f**). **c–f** large yellow dots, donor populations; small purple dots, vulnerable populations.

climate conditions than those areas where cold-dry tolerant individuals live.

**Identify potential climate-tolerant populations for evolutionary rescue.** Strikingly, while the warm-humid and warm-dry tolerant groups are likely to risk a decline in niche suitability, they are predicted to need less genetic change to cope with future climate conditions. Conversely, while the cold-dry tolerant group is likely to maintain most of its current habitats, it needs to undergo a greater genetic change. To better reflect the intraspecific variation in responses to climate change in the two species, we combined the genomic offset and NSC to estimate a genome-niche index as described in ref. 48 (see also Methods). We considered only niches with increasing suitability (positive NSC value) because these areas will remain hospitable for these species, while other areas with decreasing niche suitability (negative NSC value) will be challenged by habitat suitability decline. Under future climate conditions, the populations in the central areas of the Southwest Mountains, i.e., the junction between different ecological zones, are predicted to experience minor interruptions of genomic offset-based habitat suitability and will thus maintain their current status in the distribution ranges (Fig. 5c, d and Supplementary Fig. 7). Therefore, these populations can be considered potential donors for evolutionary rescue under climate change.

**Landscape connectivity establishes evolutionary rescue routes.** To explore how landscape features would affect the patterns of gene flow between populations, i.e., the potential for setting up evolutionary rescue routes, we implemented linear mixed-effect modelling using maximum likelihood population effects (MLPE[49]) to fit five landscape features on the matrix of pairwise genetic distance ($F_{ST}$, see Methods). We found that landscape connectivity was strongly related to a combination of habitat suitability and elevation for *P. monticolus* and habitat suitability for *T. elliotii*, supported by Akaike weights ($w_i$) and a linear mixed modelling approach (Supplementary Table 6). We then extrapolated these relationships to estimate migration possibilities from the donor populations to the populations that are potentially vulnerable to climate change. In both species, individuals from the donor populations could disperse westward to the south Tibetan zone and eastern Himalayan zone, southward to the southwest mountainous zone, northwards to the Loess Plateau zone and east meadow zone, and eastward to the western mountainous plateau zone (Fig. 5e, f). Considering that the ecological niche modelling predicted a substantial decrease in the niche suitability in the southern, northern and eastern parts of the distribution ranges, a plausible evolutionary rescue would be a westward dispersal under the assumption of a westward expansion of suitable niches even beyond the current range (Supplementary Fig. 8). Consequently, the evolutionary rescue of mountainous species depends not only on genomic offset and niche suitability estimates but also on landscape connectivity of the populations.

**Demographic model tests the migration possibility among groups.** To further explore the possibility of migration between the groups, we compared three models of gene flow (one with continuous gene flow, one with secondary gene flow and one with no gene flow, Supplementary Fig. 9) between groups using Fastsimcoal v2.6[50]. Our model tests and Akaike information criterion (AIC) comparison supported a secondary gene flow (Supplementary Table 7). The demographic parameters generated under the optimal model suggest that migration between groups is likely to have occurred between 11.5–17.5 thousand years ago in the two species (Supplementary Table 8), consistent with a possibility that Pleistocene glaciations have driven range shift and secondary contact between previously isolated groups[24–26]. The gene flow, however,

likely occurs along the areas where the three groups meet, as the mixed genotypes are mostly found in individuals from the central areas of the Southwest Mountains (Supplementary Fig. 10). Taken together, the migration estimates demonstrate that the central areas of the Southwest Mountains provide potential corridors and routes of connectivity for the three climate-tolerant groups.

## Discussion

Understanding species' responses to climate change plays a vital role in developing effective biodiversity conservation plans[51]. Intraspecific variation in the genotype-climate association has been documented for many species (e.g. refs. 12, 13, 24–26) but has only recently been considered in the vulnerability prediction to climate change[11,52]. By considering intraspecific variation in climate-associated genotypes, we show that the cold-tolerant populations have a greater genomic offset but risk less niche suitability decline under future climate conditions than warm-tolerant populations. By combining genomic offset and niche suitability change, we consider the populations with minor genome and niche interruptions to be the potential donor populations for evolutionary rescue. We then used landscape genetic analysis to identify potential rescue routes to mitigate the challenge of climate change. Altogether, our study demonstrates a framework of incorporating ecologic genomics, niche modelling and landscape genetics in predicting climate change-driven vulnerability and aiding conservation efforts.

The highly heterogeneous environments in mountainous areas often drive intraspecific genetic divergence and local adaptation of mountainous species to specific climatic conditions[24–26]. *T. elliotii* and *P. monticolus* are among the most characteristic passerines of the Sino-Himalayan Mountains. We found that the top climatic variables contributing to genomic variation in the two species were related to seasonal temperature and precipitation (Supplementary Table 2). The amplitudes of climatic variables have long been considered to play an important role in shaping the richness and biodiversity pattern of mountainous birds[18–20]. Our results demonstrate that climatic fluctuation is also the definitive factor in shaping the genomic variation and divergence of these mountainous birds, further supporting the role of topological complexity and climatic heterogeneity in shaping mountainous biodiversity[19].

By mapping genotype-climate associations across the distribution ranges of the two birds, we identified that the populations in the cold-dry areas showed distinctly different associations from those in the warm-humid and warm-dry areas. Additionally, our cluster analyses of SNPs that are significantly associated with the top climatic variables separate the cold-dry, warm-humid and warm-dry tolerant individuals. The three genetic groups are geographically congruent with the currently recognised ecological zones. For example, the cold-dry tolerant group of *T. elliotii* inhabits the southern Tibetan zone, which has an average elevation of 4500 m a.s.l. and is dominated by a plateau with a cold-dry climate and typical alpine meadow and shrubland. The warm-dry groups of *T. elliotii* and *P. monticolus* are mostly located in the western mountainous plateau zone, which mostly comprises mid-elevational mountains with a typical temperate climate. However, in the spatially heterogeneous southwest mountainous zone, we found that the populations of *T. elliotii* in the temperate coniferous forests in the northern part and the tropical broadleaf forests in the southern part are clustered into cold-dry and warm-humid tolerant groups, respectively. The cold-dry tolerant populations showed greater genomic offset values than the warm-humid tolerant populations (Supplementary Fig. 11). Altogether, the intraspecific variation in genotype-climate relationships is likely due to local adaptations to heterogeneous climatic conditions, which drive population-specific responses to future climate change.

Based on the intraspecific variation observed in the two species, we predicted population-specific genomic offsets for future climate

change. We found that the populations in the cold areas show higher genomic offsets to future climate change than the populations in the warm areas. When incorporating intraspecific variation in predictions of niche suitability under future climate conditions, we found that the niche suitability for the warm-tolerant populations will decrease to a much greater degree than for the cold-tolerant populations. In addition, we observed an expansion of the potential niches for the cold-tolerant populations but not for the warm-tolerant populations (Supplementary Fig. 8). These results suggest that warm-tolerant populations, despite having relatively minor genomic offsets to climate change, may still need to cope with habitat degradation under future climate conditions.

The cold-tolerant populations are mostly found in the mountains in the southwest mountainous zone and eastern Himalayan zone, which are characterised by drastic elevational variation from mountain tops to valleys. For example, a series of mountains, including Gongga Mountain, reach above 7000 m a.s.l., while the Red River valley lies at elevations below 300 m a.s.l.[22]. Considering that such a wide elevation gradient allows up- and down-slope dispersal, it seems reasonable to expect that cold-tolerant populations can still maintain their niche suitability even through drastic climate fluctuations. However, the warm-tolerant populations in the western mountainous plateau zone and in the southern part of the southwest mountainous zone are mostly found in the patchily distributed mid-elevational mountains. The summits of these mountains are below 4000 m a.s.l., e.g. 3771 m a.s.l. for the Qinlin Mountains (Mt. Taibai, see ref. [53]) and 3105 m a.s.l. for the Shenlongjia Mountains (Mt. Shennongding, see ref. [54]). As the species studied herein have almost reached their upper elevational limits, there are few available niches allowing uphill migration[55]. Consequently, these populations with decreasing niche suitability are more likely to be at risk for habitat decline when the climate becomes warmer.

When species are vulnerable to future climate change, their survival may depend on evolutionary rescue. The populations in the central areas of the Sino-Himalayan Mountains not only have the least genomic offset to climate changes but are also predicted to have only minor niche suitability interruption throughout climate fluctuations. Consequently, they can serve as potential donor populations for evolutionary rescue. In fact, this part of the mountains has long been considered to be 'glacial refugia', where populations were maintained throughout the Pleistocene glaciations despite the drastic climatic fluctuations, while populations elsewhere experienced severe bottlenecks[24–26].

Evolutionary rescue is also dependent on landscape connectivity between populations. The central areas of the Southwest Mountains are characterised by a series of parallel mountains, rivers and valleys running in a south-north direction[56,57], which provide corridors for the exchange of individuals living in different ecological zones[22,58]. Indeed, our landscape connectivity analyses show that dispersal possibilities are possible for southward, westward, eastward and northward migration via the central areas of the Southwest Mountains. However, as the southern, northern and eastern parts of the distribution ranges are predicted to face a substantial decrease in niche suitability, an evolutionary rescue is possible via westward migration to the southern Tibetan zone (*T. elliotii*) and the eastern Himalayan zone (*P. monticolus*), where suitable niches will expand westward. Consequently, conservation measures to preserve biodiversity from future climate change should consider not only the species' genomic offsets and niche suitability but also the landscape connectivity and their ability to disperse.

Species in the mountains often live in spatially heterogeneous environments and are thus locally adapted to different climate conditions. This intraspecific genetic variation is an important feature that should be considered when estimating vulnerability to climate change.

By considering population-level genetic variation in mountainous species in response to future climate change, we demonstrate that a combination of genomic offset and niche suitability provides unique insight into climate change-driven vulnerability. We highlight the importance of integrating multiple factors, i.e., genomic offset, niche suitability and landscape connectivity, in estimating climate change-driven vulnerability.

In reality, the actual evolutionary responses of species to climate change are more complex than those explained by the aforementioned three factors. It requires understanding interactions between local adaptation, phenotypic plasticity, evolutionary potential, effective population size, dispersal ability, and interspecific interactions (i.e., refs. [15], [51], [59]). Consequently, a combination of multiple predictors will improve our understanding of climate change-driven species vulnerability. Further implementing experiments on the physiological tolerance of species, i.e., common garden and transplant experiments[60], as well as a functional test of climate-sensitive genes[48], will shed more light on how species respond to future climate change.

## Methods

### Study areas

The distribution ranges of *T. elliotii* and *P. monticolus* cover the Sino-Himalayan Mountains (including the Southwest Mountains and East Himalayas) and a chain of mid-elevational mountains in Central China (Fig. 2a). The Southwest Mountains are located at the junction of the Palaearctic and Oriental biogeographic realms[21]. This region comprises a cluster of high mountains with a high range in elevation and heterogeneous climates. The northern part of the Southwest Mountains, the southern Tibetan zone, is located in the southeastern part of the Qinghai-Tibetan Plateau, which has an average elevation of 4500 m a.s.l. (STZ in Fig. 2a). This zone is dominated by a cold-dry climate and characterised by typical alpine meadow and shrubland. The southern part, the southwest mountainous zone (SMZ in Fig. 2a), has a vertical climatic zonation ranging from subtropical broadleaf forests and temperate coniferous forests to alpine meadows[22]. On the southwest side of the Southwest Mountains lies the East Himalayan biodiversity hotspot, which is zoo-geographically attributed to the eastern Himalayan zone (EHZ in Fig. 2a). This region is on the southern margin of the Qinghai-Tibetan Plateau and encompasses a broad range of ecological habitats varying from grassy meadows to dense humid evergreen forest. On the east side of the Southwest Mountains, a chain of mid-elevational mountains spans eastward. These mountains and adjacent lowlands belong to the western mountainous plateau zone (WMPZ in Fig. 2a).

Across their distribution ranges, *T. elliotii* and *P. monticulus* are commonly found between 1500 and 45,000 m a.s.l. Because distribution maps obtained from the International Union for Conservation of Nature (IUCN, https://www.iucn.org/) include unsuitable habitats, i.e., areas below 1000 m a.s.l., we restricted our study to areas by integrating distribution records and expert distribution maps based on ecological niche modelling[61]. This restriction removed parts of unsuitable habitats, i.e., a lowland Sichuan Basin with an average elevation of 500 m a.s.l. from our subsequent analyses.

### De novo genome assembly and annotation of *T. elliotii*

We generated a 10X Genomics linked-read library for muscle from a *T. elliotii* individual collected from Shenlongjia Mountain, Hubei Province, China (voucher number SNJ08157). Sequencing was carried out on BGI-seq 500 platform with PE150. We cleaned the raw reads using SOAPfilter v2.2 with the following steps; (1) removing reads with >10% of N; (2) removing reads with >60% low-quality bases (Phred score ≤10); (3) removing reads with undersize insert size; (4) removing PCR duplicates. All cleaned reads were used to assemble the genome using Supernova v2.0.1[62] under the 'pseudohap' mode and the intra-scaffold gaps were filled using GapCloser v1.12[63]. We measured genome

completeness using BUSCO v3.0.2[27]. We applied the homologue-based approach to annotate the protein-coding genes by using the protein sequences of *Gallus gallus* and *Taeniopygia guttata*. The protein sequences of these reference genes were aligned to *T. elliotii* genome using TABASTN v2.2.26[64] with an e-value cut off 1e⁻⁵, and multiple adjacent hits of the same query were connected by genBlastA v1.0.4[65]. Homologous blocks with lengths larger than 30% of the query protein length were retained. The connected hit region was later extended to include its 2 Kb flanking regions, on which gene structure was predicted by Genewise v2.4.1[66]. We then used Muscle v3.8.31[67] to align the annotated proteins with the reference proteins. Predicted proteins with a length ≥30 amino acids and identity value ≥40% were retained.

### Sample information for resequencing
We sampled 55 and 58 individuals from the distribution ranges of *T. elliotii* and *P. monticolus*, respectively (Fig. 2b and Supplementary Data 1). The tissues of these birds were stored in a −80 °C freezer before transportation to the sequencing centre. The Zoological Museum of Institute of Zoology has the authority for specimen collecting and exemption of export/import of samples for scientific purposes (No. 1999/84, provided by Article VII from CITES). Tissue collecting procedures conform to the regulations of the Animal Experimental and Medical Ethics Committee of the Institute of Zoology, Chinese Academy of Sciences.

### Genomic data generation and processing
Genomic DNA was extracted from tissues and subsequently sequenced on the Illumina sequencing platform (NovaSeq 6000) at Berry Genomics Corporation (Beijing, China). DNA libraries were constructed according to the manufacturer's instructions and subsequently sequenced in PE150. We conducted quality control to filter reads, (1) ≥10% unidentified nucleotides (N); (2) >10 nt aligned to the adaptor sequence, allowing ≤10% mismatches; (3) >50% bases having Phred quality <5; and (4) PCR duplicates.

Qualified reads of *P. monticolus* and *T. elliotii* were aligned against the great tit genome (GCA_001522545.3[28]) and the de novo genome of *T. elliotii* generated in this study using BWA v0.7.17[68] We classified variants using the HaplotypeCaller function from GATK v3.7.0[69] and Samtools v1.3.1[70] with default settings and then intersected two VCFs for each species to obtain the final dataset. We further used VCFtools v0.1.12b[71] and BCFtools[70] to remove indels and keep only biallelic SNPs with the following filtering criteria: (1) minQ >30, (2) min-DP >7 and max-DP <1000, (3) max-missing counts = 5, (4) SNPs at least 5 bp away from INDEL regions.

### Genomic offset modelling with gradientForest
We used gradientForest[29] to model compositional genetic turnover (i.e., turnover in allele frequencies) using nonlinear functions of climatic gradients. The turnover function transforms multidimensional climatic variables to multidimensional genetic space while selecting and weighting these variables so that they can best summarise genomic variation[11,29]. We used only SNPs with a minor allele frequency >0.05 because rare alleles tend to yield false positives[13]. We collected 19 climatic variables at 2.5-min resolution from WorldClim[72] (http://www.worldclim.org). We randomly selected 50,000 SNPs because of constraints in computational time. For each SNP, we used 500 regression trees to build a function for each of 19 climatic variables. Only SNPs with $R^2 > 0$ (a measure of the response of individual SNPs to the gradient of a given climatic variable) were considered to be 'predictive' loci and were further used in the aggregate turnover function, accounting for the importance of climatic variables and the goodness of fit for each SNP.

The gradientForest analysis provided a weighted $R^2$ value for each of 19 climatic variables to assess its importance (Supplementary Table 2). We selected the top variables by ranking their importance and discarded those that were highly correlated with a variable with a higher weighted $R^2$ value (Pearson's $r > 0.7$). We then used turnover functions to examine changes in allele frequencies among the top variables and to transform them into genetic importance values. To visualise the resulting multidimensional genetic patterns in geographic and biological space, we used principal component analysis (PCA) to reduce the transformed climatic variables into three PCs[11,29].

We used the gradientForest outputs calculated for the current climate as the baselines to predict genomic variation under future climate conditions. The Euclidean distance between the current and predicted values is referred to as the genomic offset[11]. To explore a range of potential future climate conditions, we used four climate models (CCSM4, MICRO-5, MPI-ESM-LR and CNRM-CM5-2) and four emission scenarios (RCP4.5 and RPC 8.5 for 2050 and 2070 projections, Supplementary Table 3) for a total of sixteen future climate conditions to predict genomic offset. To estimate the spatial regions where the genotype-climate relationship will be most disrupted under future climate, we transformed the climatic variables from each of the 16 climate conditions into genetic importance using the turnover functions as described above. For each grid, we calculated the Euclidean distance between the current and future genomic importance values[11,29], which serves as a metric of genomic offset. We averaged the genomic offset values of the four climate models. For each of the future climate emission scenarios and decades, we compared genomic offset values among different climate-tolerant groups using the two-tailed Wilcoxon rank-sum test. A *P* value of 0.05 after FDR correction for multiple comparisons was considered to be significant. We used two datasets, i.e., $R^2$ positive SNPs identified in the gradientForest (5446 and 7294 SNPs for *T. elliotii* and *P. monticolus*) and outlier SNPs identified in a combination of three genotype-climate association approaches (72 and 798 SNPs for *T. elliotii* and *P. monticolus*, see below), to check the consistency of the results.

### Genomic offset modelling with GDM
Ideally, the gradientForest analysis should use at least four individuals for each locality (e.g. ref. 13). However, our study areas are in highly heterogeneous mountains, which presents a great challenge for obtaining samples. Although our study, to our knowledge, represents the densest bird sampling in this region, we could only include two or three individuals of *T. elliotii* for three localities (i.e., Rangtang, Mangkang and Wulong). As these localities are on the junction areas of the three identified genetic groups, we wanted to retain them because they would help to define fine-scale population structure and estimate the extent of genetic admixture. Given this small sampling size, we implemented a deep-sequencing strategy because low precision in statistical inference due to the low sample size can be offset by using a large number of SNPs in the analyses[73,74].

We then conducted a parallel genomic offset analysis using R package GDM[46], a distance-based method (i.e., $F_{ST}$), to estimate the relationship between genetic variation and climatic variables. It has been shown that $F_{ST}$ analysis provides reliable estimates even for populations with small sample sizes (i.e., two individuals) given a large number of SNPs (i.e., >1500)[74,75]. After removing autocorrelated variables (Pearson's $r > 0.7$), we kept BIO1, BIO2, BIO4, BIO7 and BIO14 for *T. elliotii*, and BIO1, BIO2, BIO4, BIO5, BIO12 and BIO14 for *P. monticolus* in the GDM analyses. We calculated the pairwise $F_{ST}$ matrix among sampling localities based on 100,000 SNPs that were randomly extracted from genomes using diveRsity package[76], and rescaled the $F_{ST}$ values to range between 0 and 1. To estimate the relative important of these selected variables, we rescaled the maximum value of the fitted I-Splines between 0 and 1, which is proportional to variable importance. We used 'gdm.transform' to predict and map the pattern of genotype-climate association across the distribution ranges of the two species[11].

We predicted the genomic offsets to future climate conditions following the same procedure as gradientFroest analyses. Specifically, we calculated genomic offset as the extent of mismatch between current and expected future genetic variation based on genotype-climate association modelled by GDM analysis, under the sixteen climatic conditions that were obtained from four climatic models and four emission scenarios and decades (Supplementary Table 3). We predicted the genomic offset by 'predict.gdm' function, and then obtained a metric of genomic offset for each of the gridded climate points. The resultant genomic offsets were mapped with ArcGIS 10.1 to show the geographic distribution of population-level variation. We compared the genomic offset values among groups using the two-tailed Wilcoxon rank-sum test and a *P* value of 0.05 after FDR correction for multiple comparisons was considered to be significant. In addition, we run the separate GDM analyses using the datasets of 72 and 798 outlier SNPs identified for *T. elliotii* and *P. monticolus*, respectively, in the three analyses of genotype-climate associations (see below).

### Identify SNPs associated with climatic variables

To control for false positives that are often observed in individual genotype-environment programmes[77], we used three approaches to identify SNPs that are highly associated with the top climatic variables identified by gradientForest, i.e., LFMM[34], RAD and dbRAD[35,36]. For the LFMM, we ran five separate MCMC runs with a latent factor of $K = 3$ (see Results). *P* values from all five runs were combined and adjusted for multiple tests using a FDR correction of $P < 0.05$. RAD and dbRAD analyses were carried out using the R package vegan v2.5-7 (https://CRAN.R-project.org/package=vegan). Only the constrained axes with a significance of $P < 0.05$ were retained and subsequently used to build the loading value ('species scores' in vegan). SNPs were identified as outliers if their loading values were greater than three standard deviations of the average loading values. We regarded SNPs identified by all three methods as climate-associated SNPs. We annotated these SNPs using SnpEff[78] and the resultant genes were enriched using Kobas v3.0[79] with Benjamini and Hochberg adjusted *P* values.

Subsequently, we used these climate-associated SNPs to define climate-tolerant groups using a model-based clustering algorithm in Admixture v1.3[45]. We run the programme with tenfold cross-validation and 10,000 bootstrapping replicates for the coancestry cluster (*K*) between 1 and 6. The optimal cluster was selected as the one with the smallest cross-validation error[45]. The genetic ancestry of the individual is assigned to the group identified in the optimal genetic cluster ($K = 3$, see Results) if it has a proportion of inferred ancestry >60%.

### Ecological niche modelling predicted niche suitability under future climate

We used an ensemble modelling approach in the R package Biomod2[47] to model niche suitability under current and future climates. After initially including six commonly used models, i.e., maximum entropy, generalised boosted model, generalised additive model, multivariate adaptive regression splines, classification tree analysis and random forest, we combined the first four models due to the poor performance of the last two models. This ensemble framework was applied to capture the variation in different ecological niche model algorithms and enhance the robustness of prediction[80,81].

We collected distribution records from museum collections and the Global Biodiversity Information Facility (GBIF, https://www.gbif.org). We first tested how reducing sampling bias affected autocorrelation using a range of thresholds to thin distribution records, i.e., from 5 to 35 km with 5-km intervals. We used Moran's I to estimate the degree of spatial autocorrelation in each thinning threshold. Compared to that of the non-thinned records, we found that Moran's I decreased and reached a stable level at the approximately 10-km and 20-km thresholds (Supplementary Fig. 12). We decided to use a 10-km threshold to minimise sampling bias because this resolution is considered to essentially capture the environmental variation in the mountains and has been used in previous ecological niche modelling in mountainous areas (e.g., [82–84]). We removed redundant records and kept only one record every ten kilometres using the function 'thin_b' in the R package ENMwizard[85]. We run separate models for individuals that were identified as tolerant to cold-dry, warm-dry and warm-humid conditions to evaluate whether they would be affected differentially by climate change. Because some individuals of *T. elliotii* located in the adjoined areas of the three ecological zones, i.e., Mangkang, Hanyuan and Wulong, showed admixed genotypes, we removed these records from modelling. After filtering, a total of 292 records were retained for *T. elliotii* and 619 for *P. monticolus*. These records included 175 and 284 cold-dry tolerant individuals, 49 and 155 warm-humid tolerant individuals and 68 and 180 warm-dry tolerant individuals for *T. elliotii* and *P. monticolus*, respectively.

We used 19 climatic variables at a 2.5-min resolution from WorldClim[72] to model the current niche suitability. After removing autocorrelated variables (Pearson's $r > 0.7$) using usdm[86], we kept the remaining uncorrelated variables for subsequent analyses (Supplementary Table 9). We used ENMeval[87] wrapped in ENMwizard[85] with different feature classes (linear, quadratic and hinge) and regularisation multipliers (RM, from 0.5 to 5 with an increment of 0.5) and selected the best combinations for the maximum entropy models[85]. We used clamping to avoid extreme predictions for climatic values falling outside the range[88]. To enhance the foresting accuracy, we removed distribution records with the 10% lowest probability detected by ENMeval[87]. We estimated the best fitting models based on the lowest AIC[89].

For the four ecological niche models, we randomly generated 10,000 pseudo-absence points (or background points[90]) across each group's range and gave equal weights to presence data and background points (i.e., 50% balancing the weights of presence and background points to a prevalence of 0.5)[91,92]. We employed cross-validation with five repeats by randomly splitting distribution records into two subsets; 70% of the data were used to calibrate the models, and the remaining 30% were used for testing. To increase prediction accuracy, we excluded the models with AUC <0.8 or TSS <0.6 from the final ensemble prediction[92]. We assigned weights to these models based on their TSS values and constructed ensemble models by calculating the weighted mean of niche suitability across the predictions[92]. When ecological niche modelling is projected outside the range of the climatic variables on which models were calibrated, there are usually nonanalogous climates (i.e., areas where the value of at least one predictor variable is outside the training region)[93]. To minimise such uncertainties, we made conservative predictions and restricted our projections to those analogous climates that can be sampled in the current distribution ranges.

We projected niche suitability under future climate conditions using four climate models (CCSM4, MIROC5, MPI-ESM-LR and CNRM-CM5) under four emission scenarios (RCP4.5 2050, RCP8.5 2050, RCP4.5 2070 and RCP8.5 2070). We run separate models for each dataset and then averaged the projections across the four climate models. For each of the four emission scenarios and decades, we calculated the change in niche suitability between the current and future climate (NSC = niche suitability index in the future climate − niche suitability index in the current climate)[48]. A negative value indicates decreasing niche suitability, while a positive value shows increasing niche suitability in future climate conditions. We compared NSCs between the cold-dry, warm-dry and warm-humid tolerant groups in each species using a two-tailed Wilcoxon rank-sum test and an FDR-adjusted $P < 0.05$ was considered to be significant.

## Combining genomic offset and NSC

As genomic offset and NSC revealed different estimates of the climate change-driven vulnerability among the three groups, we used a genome-niche index, which allows incorporating genomic offset and NSC to better predict the responses of two species subject to future climate change[48]. Only the niches of increasing future suitability were considered because populations in these areas would not be challenged by niche suitability decline, as we aim to find the populations that are least interrupted by future climate change. The genome-niche index (gni) of each grid is calculated as follows: $gni_i = nsc_i^{\alpha} go_i^{1-\alpha}$, where $nsc_i$ and $go_i$ are the NSC and genomic offset at location $i$, respectively, and $\alpha \in [0,1]$ is the weight of normalised $nsc_i$, $i = 1, 2,..., n$. To minimise the total deviation between $ngi_i$ and $nsc_i$, $go_i$, we used the equation below to find the smallest value of $\min Y$ :[48]

$$\min Y = \sum_{i=1}^{n} \left[ \frac{\alpha \ln nsc_i + (1-\alpha) \ln go_i}{\ln nsc_i} + \frac{\ln nsc_i}{\alpha \ln nsc_i + (1-\alpha) \ln go_i} \right. $$
$$\left. + \frac{\alpha \ln nsc_i + (1-\alpha) \ln go_i}{\ln go_i} + \frac{\ln go_i}{\alpha \ln nsc_i + (1-\alpha) \ln go_i} - 4 \right]$$

We used the artificial bee colony (ABC) algorithm to find the smallest value of min $Y$ and estimate the optimal α value using the R package ABCoptim (https://CRAN.R-project.org/package=ABCoptim). We set the population size to 20, the maximum number of iterations to 1000 and the limit to 50. We normalised $nsc_i$ and $go_i$ values between 0.1 and 0.9. After the ABC algorithm converged, we obtained optimal estimates of $nsc_i$, i.e., under the projection RCP8.5 2050, $\alpha = 0.52725$ for *T. elliotii* and $\alpha = 0.617253$ for *P. monticolus*, which were further calculated to obtain $gni_i = nsc_i^{0.52725} go_i^{0.47275}$ for *T. elliotii* and $gni_i = nsc_i^{0.617253} go_i^{0.382747}$ for *P. monticolus*. We used ArcGIS 10.1 to visualise the resultant genome-niche index across each species' distribution range.

## Landscape genetic analysis

To investigate the influence of different landscape features on genetic structure, we employed a modelling approach to test a set of alternative hypotheses of landscape connectivity. We generated a resistance surface for each landscape feature, including elevation, slope, a standard deviation of elevation, habitat suitability and land cover using Circuitscape[94]. For each elevation, slope and standard deviation of elevation raster layers, we extracted values for all distribution records (used for ecological niche modelling) for each species and calculated the average value for each landscape feature. We used this value as the benchmark (i.e., the lowest resistance cost) and regarded the absolute difference between this value and the focal value of each grid as the resistance cost for this given grid. In addition, we normalised the niche suitability feature by subtracting the habitat suitability index obtained from the ecological niche modelling by 1. We rescaled these continuous values between 1 (the lowest resistance) and 10 (the highest resistance) to normalise these landscape features.

According to our field observations and written sources[23], *T. elliotii* and *P. monticolus* are mainly found in coniferous forests and broadleaf forests (and shrublands in *T. elliotii*). We thus assigned the lowest resistance (i.e., 1) to these habitat types. We then assigned a resistance cost of 3 to savannas, 5 to croplands, 7 to barren and sparsely vegetated areas and 10 to water bodies (Supplementary Table 10). Considering that this assignment is arbitrary, we carried out a sensitivity test by assigning different resistance costs to these habitat types. Our model test showed that assigning a low resistance cost to forest habitats always gave better model performance, thus validating our resistance cost assignment in land cover (Supplementary Table 10).

We randomly extracted 100,000 SNPs from the genome to calculate the genetic distance ($F_{ST}$) between populations in the R package diveRsity[76]. We used linear mixed-effect models to estimate the effects of landscape features on the patterns of gene flow in lme4[95] to account for multiple memberships by MLPE. Here, the term 'population' refers to localities. In all models, pairwise genetic distance was used as the dependent variable, and resistance costs from the landscape features were used as the independent variables. All independent variables were z-transformed to meet the normality assumption. As a high degree of collinearity between landscape features may bias the parameter estimation, we kept only features that had a Pearson's $r < 0.8$. We evaluated modelling performance by ranking AIC with second-order bias correction (AICc). The best model was selected by computing Akaike weights ($w_i$)[96]. As model selection with AICc can be biased by the nonindependence of pairwise datasets[97], we estimated model fit with marginal $R^2$ (i.e., landscape features) and conditional $R^2$ (i.e., landscape features and population effect) using MuMIn[98].

## Demographic model tests the migration possibility between groups

We used a coalescence-based approach to test the presence of migration between the groups using Fastsimcoal v2.6[50]. We first generated a two-dimensional, folded site frequency spectrum (SFS) dataset based on whole-genome-wide SNPs for each pair of groups using easySFS.py (https://github.com/isaacovercast/easySFS). We compared three demographic models for each pair of groups, M1) no occurrence of gene flow, M2) occurrence of a secondary gene flow assuming that the groups have been in secondary contact at some time, and M3) occurrence of a continuous gene flow assuming that the groups frequently exchange migrants (Supplementary Fig. 9). For each model, we run 100 replicates with 100,000 coalescent simulations with a minimum of 20 and a maximum of 40 cycles in a conditional maximisation algorithm. We specified a mutation rate of $3.3e^{-9}$ per site per generation following the estimate for passerine birds[99]. We used the AIC to evaluate which model had the higher likelihood (Supplementary Table 7). We simulated 100 replicates of the SFS from the *_maxL.par file (i.e., the parameter estimates that produced the maximum likelihood) for the best-fit run (minimised difference between maximum estimated likelihood and maximum observed likelihood) of the best-fit model. We run the 100 bootstrap replicates and ranked their maximum observed likelihoods. The demographic parameters from the top 30 runs were used to estimate confidence intervals[100].

## Reporting summary

Further information on research design is available in the Nature Research Reporting Summary linked to this article.

# Data availability

The genome assembly and sequencing data of the de novo sequenced individual of *T. elliotii* have been deposited to the National Genomics Data centre (https://db.cngb.org/) under BioProject accession number CNP0003256 and NCBI with BioProject accession number PRJNA860040. The resequencing data from *P. monticolus* and *T. elliotii* have been deposited in the National Genomics Data centre (https://db.cngb.org/) under the accession number CNP0002314 and CNP0002315, respectively. VCF datasets, location records used in ecological niche modelling, climatic data used in genotype-climate-association, outputs of ecological niche modelling and genomic offset analyses are available in Dryad (https://doi.org/10.5061/dryad.brv15dvb5).

# Code availability

Analysis scripts can be found in Dryad (https://doi.org/10.5061/dryad.brv15dvb5).

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

## Acknowledgements

We sincerely thank Huijie Qiao and Xuan Liu for their facilitating ecological niche modelling and landscape genetic analyses. This research was funded by the Strategic Priority Research Programme of the CAS (XDA20050204 to Y.Q.), the Second Tibetan Plateau Scientific Expedition and Research (STEP) programme (2019QZKK0501 to Y.Q.), and the National Natural Science Foundation of China (NSFC32020103005 to Y.Q. and 3213000355 to F.L.).

## Author contributions

Y.Q. and P.G.P.E. designed research; G.S., X.L. and F.L. collected samples; Y.C., Z.J., P.F., P.G.P.E. and Y.Q. performed research; Y.Q., Y.C. and P.G.P.E. wrote the paper. Z.J., X.L., G.S. and F.L. commented on the paper.

## Competing interests

The authors declare no competing interests.
