## [Peer Review File · Nature Communications]

REVIEWER COMMENTS

Reviewer #1 (Remarks to the Author):

This study uses the framework developed by Razgour et al. (2019, PNAS, 116, 10418–10423) to incorporate SNPs identified as associated with climatic variables into ecological niche models and then use landscape genetics to identify evolutionary rescue potential from climate adapted to mal-adapted populations. They add to this the genomic vulnerability analysis developed by Fitzpatrick & Keller (2015, Ecology Letters, 18, 1–16) and used in several studies, including Bay et al. (2018, Science, 359, 83–86) and Ruegg et al. (2018, Ecology Letters, 21, 1085–1096). Hence the approaches used are not novel, apart from the approach to identify climate adaptive populations through overlapping ecological niche models and genomic vulnerability maps. Although not entirely novel, the combination of the approaches provides new interesting insights to the analysis of climate change vulnerability. Overall the study uses a robust SNP dataset, but some of the analyses do not follow the state of the arts in the respective fields and important information for assess the quality of the models and analysis is missing.

1) Ecological niche modelling

a) In order to capture uncertainty in the modelling process due to variation in predictions of different modelling algorithms, the current standards in the field of ecological niche modelling are to use ensemble modelling approaches, which combine the outputs of different modelling algorithms (Araújo & New, 2007, Ensemble forecasting of species distributions. Trends Ecol Evol 22, 42–47; Araújo et al., 2019, Standards for distribution models in biodiversity assessments. Science Advances, 5, eaat4858).

b) Missing information on Maxent model performance, i.e. AUC scores and true skills statistics (TSS). Without this information we cannot assess whether the models are able to discriminate between presence and background points.

b) Missing information on how spatial autocorrelation and sampling bias in location records were addressed.

c) Why were the only regularization values tested when parameterising the Maxent models 0.5, 1 and 1.5? Studies commonly test also at least reg 2 and 3.

d) How were intermediate individuals (belonging to 2 or more clusters) dealt with in the ecological niche models?

2) Why was only one genotype-environment associations analysis (GEA) method used to identify SNPs associated with climatic variables? LFMM has high false positive rates, and therefore it is recommended to use it alongside another GEA method and only include SNPs that overlap between the two methods (Rellstab et al., 2015, A practical guide to environmental association analysis in landscape genomics. *Mol Ecol*, 24, 4348-4370).

3) Missing information on parameters used to run the STRUCTURE analysis - number of iterations, number of burn-in, number of replicates etc...

4) Missing justification for the choice of 20% threshold for low vulnerability (L452-461). If analysis with thresholds 20% and 10% showed similar results, why not use the more stringent threshold?

5) Landscape genetics analysis

a) I am unclear on how the landscape genetics analysis was carried out. Simple correlations between F_{st} and resistance variables are not suitable for this analysis because of lack of independence of data points. Each population is tested against several other populations. Such pairwise analysis requires a statistical method that takes this into account. The currently commonly used method in the field of landscape genetics is Maximum Likelihood Population Effect Models (Van Strien et al., 2012, A new analytical approach to landscape genetic modelling: Least-cost transect analysis and linear mixed models. *Mol Ecol*, 21, 4010–4023).

b) Missing information on how resistance costs were assigned to land cover variables. What guided the cost selection and did you test for the effect of varying the resistance costs

c) Why were continuous resistance costs maps not used? These often outperform maps divided into arbitrary 10 categories.

6) Data accessibility section missing coordinates of location records used in the ecological niche models, environmental data used in the LFMM analysis and outputs of the ecological niche models and genomic vulnerability analysis.

Reviewer #2 (Remarks to the Author):

Thank you for this interesting manuscript exploring climate change vulnerability of two Sino-Himalayan bird species using multiple modeling approaches. I found the motivation for the manuscript interesting and useful and of broad interest. The manuscript could be improved by more careful use of terminology and by providing crucial methodological detail necessary to evaluate and replicate the study.

Major Comments:

(1) I found the manuscript difficult to follow at times owing to (i) grammatical issues and inconsistent / inappropriate use of terminology and (ii) a substantial lack of important details regarding the modeling methods. In my detailed comments, I attempted to highlight where I thought there needed to be more careful use of terminology and clarification / addition details. Overall, I found it difficult to follow the methods and to fully assess the paper given these problems. For example, for the maxent modeling, there was no mention of how the models were evaluated and no measure of model performance was presented - nor were there any details regarding how the model projections were made (i.e., was clamping used or was extrapolation allowed, etc?). In general, I had a great deal of difficulty following exactly how the models were implemented and evaluated.

(2) By my reading, gradient forest was used to identify the top five most important climate variables, which were then used in LFMM to identify candidate SNPs. If that is correct, I think this could be problematic because it assumes that the primary climate gradients associated with the genome as a whole are the same as those driving climate adaptation. There are of course good reasons why we might not expect this assumption to be valid. Better / clearer justification for this approach is needed. Alternatively, it might be better to first use LFMM to identify candidate SNPs and then to fit gradient forest to these data.

(3) Much of the paper emphasizes comparisons of genomic offset predictions from gradient forest to projected losses of suitable habitat from a niche model. However, how these comparisons were made is potentially problematic. For example, to estimate habitat loss, the continuous predictions from maxent were converted to 0/1 using a threshold & these 0/1 maps were then used to calculate habitat loss in each ecoregion. This allowed pixel-by-pixel mapping of the loss / retention of suitable habitat. In contrast, the continuous genetic offsets were in essence averaged across each region. Using this method the study concluded for example that "warm-adapted" populations had lower average genomic offset than "cold-adapted" populations, but "warm-adapted" populations were expected to suffer the loss of larger areas of suitable habitat than "cold-adapted" populations. The problem is, at what magnitude of genomic offset do we expect the "loss" of a population? A more appropriate approach would be to compare the changes in continuous habitat suitability to the genomic offsets and ask how these two measures of risk are related (or not).

Other comments:

L18-20: This study did not assess “adaptive capacity” but rather the expected magnitude of maladaptation and changes in habitat suitability.

L29-30: Accuracy was not assessed in this study.

L36-37: I think most studies have shown that land use change is *currently* a more important driver of biodiversity loss than climate change. Climate change is expected to become more important in the future. The cited study by Urban talks about future projections not current trends.

L41: What is meant by “climate models” here? Does this refer to actual models of the climate system or another term for “niche model”?

L45: Other studies made the case for incorporating evolution / adaptation into niche models before Bay et al. Gotelli & Stanton-Geddes (2015) and/or Fitzpatrick & Keller (2015) are better options. Also, one could argue that Bay et al did not truly consider “adaptive capacity” per se since in that study no attempt was made to correct for population structure or to identify candidate SNPs.

L50: Please consider not using the term “genomic vulnerability” if possible. I recognize it is gaining popularity in the literature, but there is already a well established definition of climate change vulnerability (see Foden et al 2019) and “genomic vulnerability” does not align well with that definition. Also, it is not clear to what extent analyses of genomic patterns relate to actual vulnerability (versus say neutral demographic patterns that have no relationship to climate). Terms like “genomic offset” or “maladaptation” might be considered better alternates.

L53: The study by Bay et al, strictly speaking, did not assess “adaptive capacity”.

L56-58: The argument that is often made for the need to incorporate adaptive capacity into models is for species with large ranges that span broad climate gradients - in other words, species for which populations are expected to exhibit local adaptation to environments that vary a lot across their range. The argument that is presented here is precisely opposite that idea & as written it is not clear why one

would expect species that are geographically isolated, narrow-ranged, and/or specific to certain environments would be comprised of multiple locally adapted populations. Please clarify.

L95-118: Should the description of the study region be moved to the Methods? Actually, it seems that the first few paragraphs in the section labeled “Results” are actually Methods.

L140-142: Not clear why there would be an expectation that studies on entirely different species would find the same set of important climate variables.

L144-145: Does not follow from the previous text. How do the results suggest anything about sensitivity to climate change?

L154: “possess standing genetic variation associated with climate change” - not clear what this statement means. The cited figures are for current climate and do not incorporate climate change in any way.

L205-208: Genomic offset will also be a function of how much climate is expected to change. How does the magnitude of climate change differ across the different ecoregions?

L312-313: Need to be careful about terminology here. The gradient forest analyses do not reveal anything about “adaptive capacity” but rather provide a metric of “maladaptation” expected given current climate-genomic patterns and the amount of climate change in a location. Again, this is NOT the same as “adaptive capacity”, which is a measure of how much a population might be expected to reduce climate change risks via an adaptive response. In contrast, genetic offset is a measure of the amount of adaptation required to retain the status quo. In other words, adaptive capacity is the *potential* for adaptation and genetic offset is *need* for adaptation.

L314-315: I think a lot of the discussion in Supp. Note 2 is really useful & it would be better to have the most important points in the main manuscript.

L349-350: I am not sure I follow the logic here. Couldn't the vulnerable populations migrate to new areas to reduce vulnerability and thereby also contribute to evolutionary rescue in the new environment?

L407 and elsewhere: See comment above regarding use of the term “genomic vulnerability”.

L416-420: What was the spatial resolution of the climate grids?

L422: Need more details / clarification here. As I understand it, GF was used to determine which climate variables were important, and these important variables (the top five) were then used in LFMM to identify candidate SNPs. Is that correct? If so, I wonder about the risk of the GF analyses identifying climate gradients associated with genomic variation *as a whole* (including neutral / background variation) - which may not be the same as the primary gradients underlying *climate* adaptation.

L430-437: How many occurrence records were available for Maxent modeling of each of the populations?

Figure 2: What are the points in panel (b)? Clearly they are locations of each species, but are these the specimens that were used in the analyses?

Figure 4: I was confused about exactly what figure 4e and figure S3 were attempting to show in terms of range loss due to climate change. These are really important for interpreting the results, but I was at a loss as to what the shaded points were meant to show. I think that the shaded areas indicate areas of suitable climate that remained in the future in each ecoregion. Correct?

Reviewer #3 (Remarks to the Author):

The premise of this paper is that you can predict genomic vulnerability to climate change but I am not convinced you can because we don't know what genes will underpin climate change responses? Are the genes that are currently under climatic selection going to be important for contributing to climate change or could rare/neutral alleles that are currently not under selection

I also didn't understand how the authors could predict the future genomic composition that will be important for climate change. I understand they followed (12) but without having to decompose that paper what is the basic premise here?

I don't like the terminology of adaptive genetic variation when all the authors have done is associate loci with environmental variables. Whether or not these loci are indeed adaptive will depend on how they influence fitness in the wild. The authors assume because there is a signature of selection these genes are adaptive and while I don't disagree with this premise big chunks of these genes might not contribute to fitness. The only true way is to link SNP's to fitness and hence phenotypes.

Line 19 and 24 in rather than into

Line 26 I don't understand what the authors mean by climate-adaptive populations.

Line 44 The authors should cite Bush et al Ecology Letters who take estimates of additive genetic variance and hence the evolutionary potential of phenotypes and integrate these estimates into species distributional models. As far as I am aware no one has been able to link genomic signatures of selection to the evolutionary potential of an organism, which is an organisms potential to respond to selection.

Line 46 This is not an accurate definition of adaptive capacity.

Line 51 But this also assumes that there is no gene flow between populations and assumes that populations do not possess the underlying genetic variation to adapt to climate change. Perhaps they have alleles, at low frequency because they are not under strong selection but as the environment changes these alleles will rise in frequency. If any populations are "better" adapted to future climates these alleles will be passed to the vulnerable populations.

Line 56 Revise sentence

Line 60 Have studies looked at the birds you study here and looked at their capacity to respond plastically or even how the traits vary across populations? Is their great potential for plasticity, I guess you could say that in lots of birds there is evidence for plasticity in traits X,Y and Z which we think are going to be important for climate change so we make the assumption that the same is for these species but it is still an assumption.

Line 68 But here you say you know nothing about their adaptive potential.

Line 77 Is there gene flow between these populations? There is a difference between local adaptation, which you can test by common garden or reciprocal transplant experiments to look at how fitness changes, and isolated populations that no longer mix.

Line 122 Is this fairly standard number of individuals for estimating population-level variation. I noticed some populations are captured by only 2 individuals which seem low for capturing population-level variation (Table S1). Is there a link between genomic vulnerability and the number of individuals sampled? It would be good to have warm and cold-adapted environments put into Table S1.

Line 136 By genomic variations you mean SNP?

Line 143 I don't see any environmental variable related to precipitation as being amongst your 3 most important environmental variables.

Line 144 I don't understand the authors logic here, are they saying that because the environment was found to associate with SNP's these species are more sensitive to climate change? I think most organisms associate with the environment in one way or another.

Line 149 It is really difficult for me to understand the geography of your study populations and all I want to know is there gene flow between these populations?

Line 153 What do you mean here, did different environmental variables associate with species from the northern edges?

Line 154 I don't think you can say this, who know what genes will underpin climate change responses.

Line 161 I don't understand how the authors are predicting future genomic compositions and their methods didn't help me understand this either.

Line 175 Are the authors using the same measure as Fitzpatrick and Keller: genomic offset?

Line 183 Is this just based on the fact that the environment is expected to shift under climate change to a greater degree than other environments? I am failing to see what the genomic angle brings to the

table because if that is the case then simply modelling the expected change in environment would tell you the same thing.

Line 195 Do these genetic clusters match up with the environmental conditions? Could you not just say this by looking at the species distributions and environments they are found in?

Line 205 Is this not circular? You used genomics to define the groups and then use the groups to define genomic vulnerability?

Line 211 I don't quite understand the author's argument here around intraspecific variation.

Line 244 I am still wondering what is the gene flow between these populations because surely if there is gene flow then populations with low genomic vulnerability will simply mate with populations with high genomic vulnerability?

Line 263 Should a discussion of this go into the intro?

I felt like the author's results was a results/discussion and a lot of their discussion was a re-hash of their results

Line 278 Precipitation was not listed as important above? And if precipitation is important is it worth noting that our understanding of how precipitation will change in the future is less clear.

Line 281 I don't think the authors are testing theories around niche conservatism. They also have not measured thermal tolerance but are just assuming that their correlations between environment and genes translate into thermal tolerance and fitness differences.

Link 333 Did you need genomics to get to this conclusion?

Reviewers' comments:

Reviewer #1:

This study uses the framework developed by Razgour et al. (2019, PNAS, 116, 10418–10423) to incorporate SNPs identified as associated with climatic variables into ecological niche models and then use landscape genetics to identify evolutionary rescue potential from climate adapted to mal-adapted populations. They add to this the genomic vulnerability analysis developed by Fitzpatrick & Keller (2015, Ecology Letters, 18, 1–16) and used in several studies, including Bay et al. (2018, Science, 359, 83–86) and Ruegg et al. (2018, Ecology Letters, 21, 1085–1096). Hence the approaches used are not novel, apart from the approach to identify climate adaptive populations through overlapping ecological niche models and genomic vulnerability maps. Although not entirely novel, the combination of the approaches provides new interesting insights to the analysis of climate change vulnerability. Overall the study uses a robust SNP dataset, but some of the analyses do not follow the state of the arts in the respective fields and important information for assess the quality of the models and analysis is missing.

Response: We thank the reviewer for a precise summary of our work, and for constructive comments to improve our work. We have addressed each of these questions in detail below.

Major comments:

Q1. Ecological niche modelling

a) In order to capture uncertainty in the modelling process due to variation in predictions of different modelling algorithms, the current standards in the field of ecological niche modelling are to use ensemble modelling approaches, which combine the outputs of different modelling algorithms (cAraújo & New, 2007, Ensemble forecasting of species distributions. Trends Ecol Evol 22, 42–47; Araújo et al., 2019, Standards for distribution models in biodiversity assessments. Science Advances, 5, eaat4858).

Response: The reviewer raised an important issue of modeling uncertainty and suggested to use an ensemble modeling approach. We fully agree with the reviewer and have followed this comment and carried out an ensemble modeling with different modeling algorithms using Biomod2. We initially employed six commonly used models, *i.e.*, maximum entropy, generalized boosted model, generalized additive model, multivariate adaptive regression splines, classification tree analysis and random forest, and finally combined the outputs of the first four ecological niche models because the last two models had low TSS and AUC values.

Specifically, we used ENMeval (77) wrapped in ENMwizard (75) with different feature classes (linear, quadratic and hinge) and regularization multiplier (RM, from 0.5 to 5 with an increment of 0.5) and selected the best combinations for the maximum entropy models (77). We used clamping avoiding extreme predictions for climatic values falling outside the range

(78). To enhance the foresting accuracy, we removed distribution records with 10% lowest probability detected by ENMeval (77). We estimated the best fitting models based on the lowest Akaike information criterion (AIC, 79).

For the four ecological niche models, we randomly generated 10,000 pseudo-absence points (or background points, 80) across each group's range, and gave equal weights to presence data and background points (*i.e.*, 50% balancing the weights of presence and background points to a prevalence of 0.5) (81-82). We employed cross-validation with five repeats by randomly splitting the distribution records into two subsets; 70% of the data were used to calibrate the models and the remaining 30% were used for testing. In order to increase prediction accuracy, we excluded the models with AUC < 0.8 or TSS < 0.6 from the final ensemble prediction (82). We assigned weights to these models based on their TSS values and constructed ensemble models by calculating the weighted mean of niche suitability across the predictions (82).

We have added this part of description to Methods (lines 553-599). In addition, we have also cited Araújo et al., 2019 and Araújo & New, 2007 (72-73).

b) Missing information on Maxent model performance, i.e. AUC scores and true skills statistics (TSS). Without this information we cannot assess whether the models are able to discriminate between presence and background points.

Response: Thank you for pointing out missing information on model performance. We have now added AUC scores and TSS in the Results in lines 232-234, as well as provided a table (Table S7) to show model performance as below:

“All ecological niche models have greater support and discrimination ability (true skill statistics (TSS), 0.71-0.88; area under the receiver operating characteristic curve (AUC), 0.9-0.96, Table S7)”.

c) Missing information on how spatial autocorrelation and sampling bias in location records were addressed.

Response: The reviewer raised the issue of how to control spatial autocorrelation and sampling bias in the location records. In our ecological niche modeling, in order to minimize sampling bias in distribution records, we removed redundant records and only kept one record within every ten kilometer (74) using function “thin_b” in the R package ENMwizard (75). We used 19 climatic variables at 2.5-min resolution from WorldClim (64) to model current niche suitability. We removed auto-correlated climatic variables (Pearson's $r > 0.7$) using the usdm (76) and then obtained the uncorrelated climatic variables for the ecological niche modeling (Table S11). We have now added this part of information in the Methods (lines 562-564; lines 574-576). We have also provided a supplementary table to present climatic variables used in the ecological niche modeling (Table S11).

d) Why were the only regularization values tested when parameterising the Maxent models 0.5, 1 and 1.5? Studies commonly test also at least reg 2 and 3.

Response: We thank the reviewer for the suggestion to test Maxent models with regularization values at least reg 2 and 3. We have followed this suggestion and tested models with different regularization values (from 0.5 to 5 with an increment of 0.5). This part of information has been added to Methods (lines 576-579).

e) How were intermediate individuals (belonging to 2 or more clusters) dealt with in the ecological niche models?

Response: Individuals showing intermediate climate-associated genotypes are distributed in the adjoined areas where the three groups are in contact, *i.e.*, Mangkang, Hanyuan and Wulong. In our ecological niche modeling, we removed the distribution records from these areas and only based analyses on those individual records that can be confidently assigned to each of the three climate tolerant groups. This has been clarified in the Methods (lines 564-572).

Q2. Why was only one genotype-environment associations analysis (GEA) method used to identify SNPs associated with climatic variables? LFMM has high false positive rates, and therefore it is recommended to use it alongside another GEA method and only include SNPs that overlap between the two methods (Rellstab et al., 2015, A practical guide to environmental association analysis in landscape genomics. Mol Ecol, 24, 4348-4370).

Response: The reviewer raised an important issue that additional genotype-environment association (GEA) analysis is required to control for a high rate of false positives due to using single GEA method. We have followed the reviewer's suggestion and carried out two more GEA analyses (RDA and dbrDA) and considered SNPs identified by all three methods to be the climate-associated SNPs. This leads a total of 70 and 798 candidate SNPs for *T. elliotii* and *P. monticolus*, respectively. These datasets have been used for the population genetic structure and gene annotation analyses. New analyses and results have been added to Results (lines 182-188) and Methods (lines 530-541). In addition, we have also cited Rellstab et al. 2015 (69).

Q3. Missing information on parameters used to run the STRUCTURE analysis - number of iterations, number of burn-in, number of replicates etc...

Response: Thank you for pointing out the missing information in the population genetic structure analysis. We estimate genetic structure based on the climate-associated SNPs using a model-based clustering algorithm in Admixture v1.3 (44). To explore individual convergence, we pre-defined the number of genetic clusters (K) between 1 and 6, with 10-fold cross-validation (CV) and 10,000 bootstrap replications. We considered the K cluster with the smallest CV value as the optimal K value (44). The genetic ancestry of the individual is assigned to the group identified in the optimal genetic cluster ($K=3$, see Results) if it has the proportion of inferred ancestry $>60\%$. This methodological detail has been clarified in the Methods (lines 545-550).

Q4. Missing justification for the choice of 20% threshold for low vulnerability (L452-461). If analysis with thresholds 20% and 10% showed similar results, why not use the more stringent threshold?

Response: Thanks to the reviewer for raising this unclear point in the threshold selection for low vulnerability estimate. In our previous version of manuscript, we wanted to show a gradient of climate-change driven vulnerability, and then used a least (10%) threshold and a minor (20%) threshold. We have now realized that these threshold selections were arbitrary and redundant.

In the revised version of manuscript, we have calculated a genome-niche index, which allows incorporating genomic offset and niche suitability change (niche suitability index in future climate - niche suitability index in current climate) for predicting the responses of two species subject to future climate change (46). We implemented a linear normalization transfer function on genomic offset and niche suitability change values and then used the artificial bee colony (ABC) algorithm to find the smallest value of $\min Y$ to minimize the total deviation between genome-niche index (gni), niche suitability change (nsc) and genomic offset (go) values, and estimate the optimal α values. After ABC algorithm converged at the 1,000 iterations, we obtained an optimal estimate of nsc_i , $\alpha=0.370556$ for *T. elliotii* and $\alpha=0.524022$ for *P. monticolus*, which were further calculated to get $gni_i = nsc_i^{0.370556} go_i^{0.629444}$ for *T. elliotii* and $gni_i = nsc_i^{0.524022} go_i^{0.475918}$ for *P. monticolus*. We used ArcGIS 10.1 to visualize the resultant genome-niche index across each species' distribution range. We have added this part of analysis to Results (lines 255-265) and Methods (lines 613-633).

Q5. Landscape genetics analysis

a) I am unclear on how the landscape genetics analysis was carried out. Simple correlations between *Fst* and resistance variables are not suitable for this analysis because of lack of independence of data points. Each population is tested against several other populations. Such pairwise analysis requires a statistical method that takes this into account. The currently commonly used method in the field of landscape genetics is Maximum Likelihood Population Effect Models (Van Strien et al., 2012, A new analytical approach to landscape genetic modelling: Least-cost transect analysis and linear mixed models. *Mol Ecol*, 21, 4010–4023).

Response: The reviewer raised an important issue that the non-independence of pairwise comparison should be taken into account in landscape genetic analysis. In the revised version of the manuscript we have followed the reviewer's comment and carried out a new landscape genetic analysis. Specifically, we used the linear mixed-effect models to estimate the effects of landscape features in the patterns of gene flow in lme4 R package (86), to account for multiple memberships with maximum likelihood population effect (MLPE). In all models, the pairwise genetic distance was used as the dependent variables and resistance costs from the landscape features as the independent variables. We evaluated the relative support for each model by ranking the Akaike's information criterion with second-order bias correction (AICc).

The best model was selected by computing Akaike weights (w_i) for each candidate model (87). As model selection with AICc may be biased by the non-independence of pairwise distance data (88), we also estimated the model fit for generalized mixed-effect models with marginal R^2 (*i.e.* landscape features) and conditional R^2 (*i.e.* landscape features and population effect) using R package MuMIn.

We have added this part of new analysis to the Results (lines 268-276) and Methods (lines 659-672). We have also cited Van Strien et al. 2012 (88).

b) Missing information on how resistance costs were assigned to land cover variables. What guided the cost selection and did you test for the effect of varying the resistance costs

Response: The reviewer pointed out the missing information on how resistance costs were assigned to land cover variables. According to our field observation and written sources (23), *T. elliotii* and *P. monticolus* are mainly found in coniferous forests and broadleaf forests (and shrublands in *T. elliotii*), we thus assigned the lowest resistance (1) to these habitat types. We then assigned a resistance cost of 3 to savannas, 5 to croplands, 7 to barren and sparsely vegetation, and 10 to water body (Table S12). As the reviewer has pointed out, we realized that this assignment is arbitrary, and we have thus carried out a sensitivity test by assigning different resistance costs to the land cover variables. Our model test showed that assigning a low resistance cost to forest habitats always gave the better model performance, which thus validated our resistance cost assignment in land cover (Table S12). We have added this part of texts to the Methods (lines 650-657), and provided a table (Table S12) to present model comparison.

c) Why were continuous resistance costs maps not used? These often outperform maps divided into arbitrary 10 categories.

Response: We agree with the reviewer that continuous resistance costs outperform arbitrary ten categories. We have thus followed the reviewer's suggestion and used continuous variables in the landscape genetic analysis. Briefly, for each of the elevation, slope and standard deviation of elevation raster layers, we extracted values for all distribution records for each species (used for ecological niche modeling). We set the average value to be the lowest resistance cost (provided as the baseline). Using this value as the benchmark, we calculated the absolute difference between this value and the focal value of each grid and then regarded the difference as the resistance cost for this given grid. We then rescaled these continuous values between 1 (the lowest resistance) and 10 (the highest resistance) to normalize these landscape features. We have added this part of information to the Methods (lines 636-648).

6) Data accessibility section missing coordinates of location records used in the ecological niche models, environmental data used in the LFMM analysis and outputs of the ecological niche models and genomic vulnerability analysis.

Response: We have now deposited these datasets in Dryad (doi.org/10.5061/dryad.br15dvb5) and clarified in Data accessibility (lines 677-680).

Reviewer #2 (Remarks to the Author):

Thank you for this interesting manuscript exploring climate change vulnerability of two Sino-Himalayan bird species using multiple modeling approaches. I found the motivation for the manuscript interesting and useful and of broad interest. The manuscript could be improved by more careful use of terminology and by providing crucial methodological detail necessary to evaluate and replicate the study.

Response: Thank you for the encouraging summary of our work, and many constructive comments to improve our work. We have followed your comments and substantially revised the previous version of the manuscript. We have addressed each question in detail below.

Major Comments:

Q1. I found the manuscript difficult to follow at times owing to (i) grammatical issues and inconsistent / inappropriate use of terminology and (ii) a substantial lack of important details regarding the modeling methods. In my detailed comments, I attempted to highlight where I thought there needed to be more careful use of terminology and clarification / addition details. Overall, I found it difficult to follow the methods and to fully assess the paper given these problems. For example, for the maxent modeling, there was no mention of how the models were evaluated and no measure of model performance was presented - nor were there any details regarding how the model projections were made (i.e., was clamping used or was extrapolation allowed, etc?). In general, I had a great deal of difficulty following exactly how the models were implemented and evaluated.

Response: The reviewer raised two important issues and we would like to address them one by one.

- a) The first issue is inappropriate use of terminology, for example, “genomic vulnerability” and “adaptive capacity”. The reviewer has also suggested a review paper of Foden et al. 2019 for terminology usage (R2Q9). We have read Foden et al. 2019 carefully and realized that the inappropriate use terminology in the previous version of manuscript has caused confusion. We have thus revised the terminology in the revised manuscript accordingly to Foden et al. 2019. For example, we have changed to “genomic offset” instead of the previously used “genomic vulnerability”. We have removed the term of “adaptive capacity” as our study does not investigate “adaptive capacity”, instead, we focus on how intraspecific variation in genotype-climate association impacts genomic offset and niche suitability change under future climate conditions. We have thus changed “adaptive capacity” to “intraspecific variation in genotype-climate association” in the revised manuscript.
- b) The second issue is the lack of details in the description of the ecological niche modeling method. In the revised version of manuscript, we have reanalyzed the ecological niche

modeling by implementing an ensemble modeling approach (as suggested by the reviewer 1). We have paid a close attention to describe the methods.

Specifically, we used an ensemble modeling approach in the R package *Biomod2* (45) to model niche suitability under current and future climate conditions. After initially including six commonly used models, *i.e.*, maximum entropy, generalized boosted model, generalized additive model, multivariate adaptive regression splines, classification tree analysis and random forest, we combined the outputs from the first four models (the last two models were excluded due to poor performance).

We used 19 climatic variables at 2.5-min resolution from *WorldClim* (64) to model current niche suitability. After removing auto-correlated variables (Pearson's $r > 0.7$) using the *usdm* (76), we kept the uncorrelated variables for subsequent analyses (Table S11). We used *ENMeval* (77) wrapped in *ENMwizard* (75) with different feature classes (linear, quadratic and hinge) and regularization multiplier (RM, from 0.5 to 5 with an increment of 0.5) and selected the best combinations for the maximum entropy models (77). We used clamping to avoid extreme predictions for climatic values falling outside the range (78). To enhance the foresting accuracy, we removed the distribution records with the 10% lowest probability detected by *ENMeval* (77). We estimated the best fitting models based on the lowest Akaike information criterion (AIC, 79).

For the four ecological niche models, we randomly generated 10,000 pseudo-absence points (or background points, 80) across each group's range, and gave equal weights to presence data and background points (*i.e.*, 50% balancing the weights of presence and background points to a prevalence of 0.5) (81-82). We employed cross-validation with five repeats by randomly splitting distribution records into two subsets; 70% of the data were used to calibrate the models and the remaining 30% were used for testing. In order to increase prediction accuracy, we excluded the models with $AUC < 0.8$ or $TSS < 0.6$ from the final ensemble prediction (82). We assigned weights to these models based on their TSS values and constructed ensemble models by calculating the weighted mean of niche suitability across the predictions (82).

When ecological niche modeling is projected outside the range of the climatic variables on which models were calibrated, there are usually non-analogous climates (*i.e.*, areas where the value of at least one predictor variable is outside the training region) (83). In order to minimize such uncertainties, we made conservative predictions and restricted our model projections onto those analogous climates that can be sampled by distribution and background records in the current distribution ranges.

The predicted ecological niche models have great support and discrimination ability (true skill statistics, TSS, 0.71-0.88; area under the receiver operating characteristic curve, AUC, 0.9-0.96, Table S7).

We have added this part of text in the Methods (lines 553-599), and updated Results (line 232-234).

Q2. By my reading, gradient forest was used to identify the top five most important climate variables, which were then used in LFMM to identify candidate SNPs. If that is correct, I think this could be problematic because it assumes that the primary climate gradients associated with the genome as a whole are the same as those driving climate adaptation. There are of course good reasons why we might not expect this assumption to be valid. Better / clearer justification for this approach is needed. Alternatively, it might be better to first use LFMM to identify candidate SNPs and then to fit gradient forest to these data.

Response: Thanks to the reviewer for raising this unclear point in our gradientForest and LFMM analyses. We realized that some details that were missing in the two analyses have brought confusion.

We used gradientForest to not only identify the top climatic variables but also genomic variants that are significantly associated with the gradients of these top variables. Specifically, we randomly selected 50,000 SNPs because of constraints in computational time, and for each SNP we used 500 regression trees to build a function for each of the 20 climatic variables. Only SNPs with $R^2 > 0$ (R^2 is a measure of response of individual SNP to gradient of a given climatic variable) were considered to be ‘predictive’ loci and were used in the aggregate turnover function, accounting for importance of climatic variables and the goodness of fit for each SNP. Out of 50,000 randomly extracted SNPs, we identified 5,476 and 7,354 SNPs that were significantly associated with the top climatic variables for *T. elliotii* and *P. monticolus*, respectively. Using an aggregate community-level turnover function across these SNPs, we visualized the genotype-climate turnover surface across the distribution ranges of the two species. We then used this gradientForest output based on the current climatic condition as the baseline to predict genomic variation under future climate condition. The Euclidean distance between the current and predicted values is regarded as genomic offset (11). We have clarified this part of information in the Methods (lines 472-506).

As the gradientForest analysis showed different genotype-climate associations and genomic offsets across the distribution ranges of the two species, we wanted to know how this intraspecific variation has driven population-specific responses to climate change. In order to address this, we used LFMM to identify SNPs that are significantly associated with climatic variables as this method takes the population genetic structure into account, thus controlling for neutral divergence resulting from geographic separation. To further control for false positives, we have followed the reviewer 1’s comment (R1Q2) to carry out two more genotype-climate association analyses (RAD and dbRAD) and selected those SNPs identified by all three methods as the climate-associated SNPs. We used these SNPs to run a cluster analysis to define the climate tolerant groups, and then investigate whether these groups have different genomic offsets and niche suitability change under the future climate conditions. We have clarified this part of information in the Methods (lines 530-541).

The reviewer also raised an interesting issue whether it is possible to first run LFMM analysis and then carry out gradientForest analysis with the identified SNPs. It is a great idea, but it requires a pre-defined dataset. LFMM, RAD and dbRAD are computational efficient so that

they can carry out a genome-wide association, but only based on a single climatic variable at the time. We have refrained from trying this given that calculating genotype association for all 20 variables individually is quite labor-intensive, and would result in a high rate of false positives due to auto-correlation among the climatic variables. Instead, we have carried out the parallel gradientForest and GDM analyses using datasets of SNP previously found to be associated with the top climatic variables (70 and 798 SNPs identified by LFMM, RAD and dbRAD in *T. elliotii* and *P. monticolus*, respectively). The new analyses show similar results as the analyses based on the datasets of 50,000 randomly selected SNPs, suggesting that our genomic offset results are robust to different datasets. We have added this part of analysis to Results (lines 220-223) and also presented a new supplementary figure (Fig. S4).

Q3. Much of the paper emphasizes comparisons of genomic offset predictions from gradient forest to projected losses of suitable habitat from a niche model. However, how these comparisons were made is potentially problematic. For example, to estimate habitat loss, the continuous predictions from maxent were converted to 0/1 using a threshold & these 0/1 maps were then used to calculate habitat loss in each ecoregion. This allowed pixel-by-pixel mapping of the loss / retention of suitable habitat. In contrast, the continuous genetic offsets were in essence averaged across each region. Using this method the study concluded for example that “warm-adapted” populations had lower average genomic offset than “cold-adapted” populations, but “warm-adapted” populations were expected to suffer the loss of larger areas of suitable habitat than “cold-adapted” populations. The problem is, at what magnitude of genomic offset do we expect the “loss” of a population? A more appropriate approach would be to compare the changes in continuous habitat suitability to the genomic offsets and ask how these two measures of risk are related (or not).

Response: The reviewer raised an important issue on the comparisons of genomic offset and niche suitability prediction. In the previous version of manuscript, we used continuous values for the genomic offsets and absence/presence values for niche suitability. We fully agree with the reviewer that we should use the continuous values for niche suitability in order to compare with genomic offset values. In the revised manuscript, we have calculated the change in niche suitability as the difference between the current and projected future climate (NSC=niche suitability index in future climate - niche suitability index in current climate). A negative NSC value indicates a predicted decrease in niche suitability compared to current climate, while a positive NSC value shows an increase in niches suitability. We found that the NSC decreases to a much smaller degree in those areas where harbor the cold-dry tolerant groups (median NSC, *T.elliotii*, -0.007; *P. monticolus*, -0.059) than in the areas where keep the warm-dry groups (median NSC, *T.elliotii*, -0.032; *P. monticolus*, -0.125, Wilcoxon test, $P<0.001$, Fig.5c-d) and warm-humid groups (median NSC, *T.elliotii*, -0.08; *P. monticolus*, -0.112, Wilcoxon test, $P<0.001$, Fig.5a-b). This part of the analysis has been added to Results (lines 236-249) and Methods (lines 601-610).

As the genomic offset provides a measure of mismatch in genotype-climate association between current and potential future climates, the niche suitability change serves as a measure of the difference in niche suitability between current and future ecological niches, making direct comparison possible. We then incorporated the genomic offset and NSC to generate a

genome-niche index as described in Ref. (46) for predicting the responses of the two species subjected to future climate condition. We only considered the niches with positive NSC values because populations in these areas wouldn't be challenged by habitat suitability decline, as our aim is to find the populations that are least interrupted by climate change. The genome-niche index (gni) of each grid is calculated as follows: $gni_i = nsc_i^\alpha go_i^{1-\alpha}$, where nsc_i and go_i is the NSC and genomic offset at location i , $\alpha \in [0,1]$ is the weight of normalized nsc_i , $i = 1, 2, \dots, n$. To minimize the total deviation between gni_i and nsc_i , go_i , we used following equation to find the smallest value of $\min Y$ (46):

$$\min Y = \sum_{i=1}^n \left[\frac{\alpha \ln nsc_i + (1 - \alpha) \ln go_i}{\ln nsc_i} + \frac{\ln nsc_i}{\alpha \ln nsc_i + (1 - \alpha) \ln go_i} + \frac{\alpha \ln nsc_i + (1 - \alpha) \ln go_i}{\ln go_i} + \frac{\ln go_i}{\alpha \ln nsc_i + (1 - \alpha) \ln go_i} - 4 \right]$$

We used the artificial bee colony (ABC) algorithm to find the smallest value of $\min Y$, and estimate the optimal α value. After the ABC algorithm had converged, we obtained an optimal estimate of nsc_i , $\alpha=0.370556$ for *T. elliotii* and $\alpha=0.524022$ for *P. monticolus*, which were further calculated to get $gni_i = nsc_i^{0.370556} go_i^{0.629444}$ for *T. elliotii* and $gni_i = nsc_i^{0.524022} go_i^{0.475918}$ for *P. monticolus*. We used ArcGIS 10.1 to visualize the resultant genome-niche index across each species' distribution range. We have added this part of texts to Results (lines 255-265) and Methods (lines 613-633).

Minor comments:

Q4. L18-20: This study did not assess “adaptive capacity” but rather the expected magnitude of maladaptation and changes in habitat suitability.

Response: The reviewer pointed out our inappropriate use of the term “adaptive capacity” in the previous version of manuscript. We agree with the reviewer and have revised this sentence as below.

“Populations locally adapted to spatially heterogeneous environments may respond climate change differentially, but this intraspecific variation has rarely been considered in modeling vulnerability under climate change” in Abstract (lines 19-22).

Q5. L29-30: Accuracy was not assessed in this study.

Response: We have changed this sentence as below.

“Overall, we demonstrate that the integration of the genomic offset, niche suitability modeling, and landscape connectivity can improve climate-change driven vulnerability assessments and facilitate effective conservation management” in Abstract (lines 30-32).

Q6. L36-37: *I think most studies have shown that land use change is *currently* a more important driver of biodiversity loss than climate change. Climate change is expected to become more important in the future. The cited study by Urban talks about future projections not current trends.*

Response: We have now changed this sentence as “Anthropogenic climate change is one of the primary drivers of environmental change and global biodiversity loss (1-2)” in Introduction (lines 37-38). We have also replaced Urban 2015 by Wiens 2016 (Wiens, JJ. 2016. Climate-related local extinctions are already widespread among plant and animal species. PLoS Biol., 14, e2001104).

Q7. L41: *what is meant by “climate models” here? Does this refer to actual models of the climate system or another term for “niche model”?*

Response: Thank you for pointing out this unclear part in our manuscript. When we mentioned “climate models” we meant to project future climate suitability. We have clarified this as follows.

“Modeling on changes in the distribution range, suitable climatic conditions and vegetation types of species under different climate scenarios has provided considerable insights into the impacts of climate change on biodiversity (*e.g.*, 6-10)” in Introduction (lines 42-44).

Q8. L45: *Other studies made the case for incorporating evolution / adaptation into niche models before Bay et al. Gotelli & Stanton-Geddes (2015) and/or Fitzpatrick & Keller (2015) are better options. Also, one could argue that Bay et al did not truly consider “adaptive capacity” per se since in that study no attempt was made to correct for population structure or to identify candidate SNPs.*

Response: Thank you for suggesting these interesting papers. We agree with the reviewer that intraspecific variation in genotype-climate association is not as same as the “adaptive potential”. We have rephrased this sentence as below. In addition, we have also cited Gotelli & Stanton-Geddes (2015) and Fitzpatrick & Keller (2015) (11-12).

“However, as these models rely solely on the abiotic and biotic environmental changes, the ecologic genomics, which determines how the genomic variants of a species vary along current environmental gradients and how much genetic change has been required to keep up with the climate-change driven environmental changes, has only recently been integrated into modeling species response to climate changes (11-13)” in Introduction in lines 44-49.

Q9. L50: *Please consider not using the term “genomic vulnerability” if possible. I recognize it is gaining popularity in the literature, but there is already a well established definition of climate change vulnerability (see Foden et al 2019) and “genomic vulnerability” does not align well with that definition. Also, it is not clear to what extent analyses of genomic patterns relate to actual vulnerability (versus say neutral demographic*

patterns that have no relationship to climate). Terms like “genomic offset” or “maladaptation” might be considered better alternates.

Response: Thank you for suggesting “genomic offset” and recommended Foden et al. 2019. We have read the paper carefully and realized that “genomic vulnerability” is not the suitable term and have thus changed “genome vulnerability” to “genomic offset” here and after. We have also cited Foden et al. 2019 (48).

Q10. L53: *The study by Bay et al, strictly speaking, did not assess “adaptive capacity”.*

Response: We agree with the reviewer’s point and have removed “adaptive capacity” throughout the manuscript.

Q11. L56-58: *The argument that is often made for the need to incorporate adaptive capacity into models is for species with large ranges that span broad climate gradients - in other words, species for which populations are expected to exhibit local adaptation to environments that vary a lot across their range. The argument that is presented here is precisely opposite that idea & as written it is not clear why one would expect species that are geographically isolated, narrow-ranged, and/or specific to certain environments would be comprised of multiple locally adapted populations. Please clarify.*

Response: Thanks the reviewer for pointing this out. We have clarified why intraspecific genetic variation and local adaptation should be expected in the mountainous species as follows.

“Mountainous areas harbor an exceptional biodiversity and endemism but are highly vulnerable to climate change (18). This is because that the complex topography within a rather small geographical area leads to dramatic ecological stratification in the mountains. Species are often restricted into spatially heterogeneous environments and locally adapted to diversified climate conditions (18-20). These population-specific adaptations likely drive different responses under climate change because the populations likely track their own optimally environmental conditions (20). Despite a great potential, this intraspecific variation has not been considered into vulnerability estimates driven by climate change (12-13)” in lines 57-64.

Q12. L95-118: *Should the description of the study region be moved to the Methods? actually, it seems that the first few paragraphs in the section labeled “Results” are actually Methods.*

Response: We have removed the description of the study region to Methods (lines 425-441).

Q13. L140-142: *Not clear why there would be an expectation that studies on entirely different species would find the same set of important climate variables. L144-145: Does not follow from the previous text. How do the results suggest anything about sensitivity to climate change?*

Response: As the mountainous and desert species cope with large seasonal changes in temperature and precipitation, we assumed same set of the climate variables, *i.e.*, temperature

annual range, would contribute to genomic adaptation of these species. We agree with the reviewer that this inference is speculative so we have removed it from the revised version of manuscript.

Q14. L154: “possess standing genetic variation associated with climate change” - not clear what this statement means. The cited figures are for current climate and do not incorporate climate change in any way.

Response: We have changed this sentence as “These results suggest that the two species show intraspecific variation in their genotype-climate associations, which are likely subject to local adaptation to heterogeneous climatic conditions” in Results (lines 148-150).

Q15. L205-208: Genomic offset will also be a function of how much climate is expected to change. How does the magnitude of climate change differ across the different ecoregions?

Response: The reviewer asked whether the magnitude of climate change differs across the ecoregions. In order to address this issue, we compared the climate change between the different ecological niches occupied by the cold-dry, warm-humid and warm-dry tolerant populations, respectively. We calculated the difference between future and current climatic variable values and then compared those differences between the three groups. A positive value shows an increase and a negative value shows a decrease in variable change. Our comparisons show complex patterns of climate change under future climate conditions for the two species (Figure 1), because the changes differ between variables and between emission scenarios (2050RCP4.5, 2050RCP8.5, 2070RCP4.5 and 2070 RCP8.5). It is thus not possible to find a congruent pattern between genomic offsets and the magnitudes of climate change across the three climate-associated groups. We have thus refrained to further discussing this issue.

Figure 1. Differences of climatic variables under current and future climate conditions among the three climate-tolerant groups. The ranges of these values are scaled by dividing by one standard deviation.

Q16. L312-313: *Need to be careful about terminology here. The gradient forest analyses do not reveal anything about “adaptive capacity” but rather provide a metric of “maladaptation” expected given current climate-genomic patterns and the amount of climate change in a location. Again, this is NOT the same as “adaptive capacity”, which is a measure of how much a population might be expected to reduce climate change risks via an adaptive response. In contrast, genetic offset is a measure of the amount of adaptation required to retain the status quo. In other words, adaptive capacity is the *potential* for adaptation and genetic offset is *need* for adaptation.*

Response: Thank you for clarifying the terminology. We have realized that we inappropriately used the term of “adaptive capacity” in the previous version of manuscript. An appropriate term should be the “genomic offset” that measures of how much the amount of genomic change is required to match climate change. We have removed “adaptive capacity” and used “genomic offset” in the revised version of manuscript.

Q17. L314-315: *I think a lot of the discussion in Supp. Note 2 is really useful & it would be better to have the most important points in the main manuscript.*

Response: We have now taken Supp. Note 2 back to the Discussion (lines 364-378).

Q18. L349-350: *I am not sure I follow the logic here. Couldn't the vulnerable populations migrate to new areas to reduce vulnerability and thereby also contribute to evolutionary rescue in the new environment?*

Response: Thank you for raising this unclear point. What we wanted to explain is that the populations with minor genomic offset and niche suitability interruptions can be considered to be the potential donor populations for evolutionary rescue, because populations either living in areas experiencing decreased niche suitability or suffering large genomic offset need to harness more adaptation to cope with the environmental change under the future climate. We have clarified this as follows.

“The populations in the central areas of the Sino-Himalayan Mountains not only have the least genomic offset to climate changes, but are also predicted to have only minor niche suitability interruption throughout climate fluctuations. Consequently, they can serve as the potential donor populations for evolutionary rescue” in lines 382-385.

Q19. L407 and elsewhere: *See comment above regarding use of the term “genomic vulnerability”.*

Response: We have changed to “genomic offset” here and elsewhere.

Q20. L416-420: *What was the spatial resolution of the climate grids?*

Response: We collected 19 climate variables at 2.5-min resolution from WorldClim (64) (<http://www.worldclim.org>). This information has been added to Methods (line 477).

Q21. L422: *Need more details / clarification here. As I understand it, GF was used to determine which climate variables were important, and these important variables (the top five) were then used in LFMM to identify candidate SNPs. Is that correct? If so, I wonder about the risk of the GF analyses identifying climate gradients associated with genomic variation *as a whole* (including neutral / background variation) - which may not be the same as the primary gradients underlying *climate* adaptation.*

Response: The reviewer raised an important issue about missing details in gradientForest method, in particular how gradientForest identifies genomic variants associated with the climatic variables. We used gradientForest (28) to not only identify the important climatic variables, but also the genomic variants that are significantly associated with these top climatic variables. Specifically, we randomly selected 50,000 SNPs because of constraints in the computational time, and for each SNP we used 500 regression trees to build a function for each of the 20 climatic variables. Only SNPs with $R^2 > 0$ (a measure of response of individual SNP to gradient of given climatic variable) were considered to be ‘predictive’ loci and were further used in the aggregate turnover function.

The gradientForest analysis provided a weighted R^2 value for each of 20 climatic variables to assess its importance (Table S4). We selected the top variables by ranking their relative importance and discarded those that were highly correlated to a variable that ranked higher (Pearson’s $r > 0.7$). We identified 5,476 and 7,354 SNPs that were significantly associated with these top climatic variables ($R^2 > 0$) for *T. elliotii* and *P. monticolus*, respectively. Based on these SNPs, we transformed change in allele frequencies along the top climatic variables into genetic importance values. We used the gradientForest outputs under the current climate as the baselines to predict genomic variation under future climate conditions. For each grid, we calculated the Euclidean distance between the current and future genetic importance values (11, 28), which serves as a measure of the genetic offset. We have added this information to gradientForest method (lines 472-506).

Because gradientForest did not consider genetic divergence resultant from population genetic structure, we used LFMM to identify SNPs that are significantly associated with the top climatic variables with a latent factor of $K=3$ (in order to control for population genetic structure). In addition, LFMM is computational efficiency so that it can carry out genome-wide association. To further control for false positives, we have carried out two more genotype-climate association analyses (RAD and dbRAD) and regarded only the SNPs identified by all three methods as the climate-associated SNPs. We then run population genetic structure analysis based on these SNPs to define the climate-tolerant groups, and further compared whether the three groups show different degrees of genomic offsets and niche suitability change under the future climate conditions. We have clarified this part of information in Methods (lines 530-550).

Q22. L430-437: *How many occurrence records were available for Maxent modeling of each of the populations?*

Response: We have added this part of information to Methods as below.

“A total of 292 and 619 records were retained for *T. elliotii* and *P. monticolus*, which included 175 and 284 cold-dry tolerant individuals, 49 and 155 warm-humid tolerant individuals and 68 and 180 warm-dry tolerant individuals for *T. elliotii* and *P. monticolus*, respectively” in Methods (lines 569-572).

Q23. Figure 2: What are the points in panel (b)? Clearly they are locations of each species, but are these the specimens that were used in the analyses?

Response: Yellow and blue dots in Figure 2b show the sampling localities of *P. monticolus* and *T. elliotii* used in genomic analysis. We have clarified this in the figure legend.

Q24. Figure 4: I was confused about exactly what figure 4e and figure S3 were attempting to show in terms of range loss due to climate change. These are really important for interpreting the results, but I was at a loss as to what the shaded points were meant to show. I think that the shaded areas indicate areas of suitable climate that remained in the future in each ecoregion. Correct?

Response: It is correct that the shades in Figure 4e and S3 in the previous version of the manuscript were used to show remained suitable niches in the future climate conditions. Figure 4e showed the ecological niche modeling outputs from means across sixteen climatic conditions (including four climate model, MPI-ESM-LR, CCSM4, MICRO-5 and CNRM-CM5-2 under four emission scenarios, 2050 RCP4.5, 2050RCP8.5, 2070 RCP4.5, 2070). This was used to demonstrate that the cold-dry tolerant groups lose fewer suitable niches (blue shade) than the warm-humid (yellow shade) and warm-dry (green shade) tolerant groups under the future climate conditions. Four panels in Figure S3 showed the niche changes under each of 2050 RCP4.5, 2050RCP8.5, 2070 RCP4.5 and 2070RCP8.5 emission scenarios, respectively. This figure was provided to show a magnitude-dependent niche loss from the moderate scenario (RCP 4.5) to the worst scenario (RCP 8.5).

In the revised version of manuscript, we calculated the change in the niche suitability between projected future climate and current climate ($NSC = \text{niche suitability index in future climate} - \text{niche suitability index in current climate}$), and compared NSCs between the cold-dry, warm-dry and warm-humid tolerant groups in each species using Wilcoxon tests and a $P < 0.05$ is considered to be significant. We have updated the figure to show niche suitability changes among the three groups and the statistical comparisons (Figure 5a-b), as well as the niche suitability under the current and future climate conditions (Figure S5). Consequently, previous Figure 4e and Figure S3 no longer exist in the revised manuscript.

Reviewer #3 (Remarks to the Author):

Major comments

Q1. The premise of this paper is that you can predict genomic vulnerability to climate change but I am not convinced you can because we don't know what genes will underpin climate change responses? Are the genes that are currently under climatic selection going

to be important for contributing to climate change or could rare/neutral alleles that are currently not under selection

Response: The reviewer raised the issue of how our study can predict climate-change driven vulnerability considering that we don't know what genes underpin the response to climate change. This made us realize that some information about method used to predict the genomic offset was missing in the previous version of manuscript, which may have caused a logic gap. In our study, we used the gradientForest to predict genomic offset under future climate conditions. The gradientForest doesn't depend on the known genes that may be relevant to climatic change responses. Instead, it detects statistical associations between the genetic variation in the studied populations based on allele frequencies of genetic variants (*i.e.*, SNPs) and the gradients of the climatic variables. Based on predictive SNPs that are identified to significantly associate with the climatic variables, the gradientForest models an aggregate turnover function for genotype-climate association and predicts genomic offsets as the mismatch in genotype-climate associations between current and predicted future climate.

The reviewer also asked whether genes that currently are under climatic selection or non-selected rare/neutral alleles will contribute to future climate change. It should be noted that the gradientForest uses all the climate-associated loci to build the functions for genotype-climate associations and doesn't distinguish the loci in the genic regions or the neutral regions. This seems a reasonable assumption because 1) causal genes or alleles underpin climate adaptation are rarely known because climate adaptation is a complex process and may involve many polygenic traits (Foden et al. 2019; Rose et al. 2018), 2) selectively neutral loci may link with selective loci so that it is hard to distinguish them. This genotype-climate association analysis is a currently well-established method (*e.g.*, gradientForest and Generalized dissimilarity modeling) and increasingly used in modeling climate-change driven vulnerability in the ecological genomics (*e.g.*, Bay et al. 2018; Ruegg et al, 2018; Dauphin et al. 2020; Rhon  t et al. 2020; Wood et al. 2021; Smith et al. 2021; Chen et al. 2021). We have elaborated on the methodological details in Introduction (lines 45-49) and Methods (lines 472-506).

Q2. I also didn't understand how the authors could predict the future genomic composition that will be important for climate change. I understand they followed (12) but without having to decompose that paper what is the basic premise here?

Response: The reviewer raised an important issue about the method (*i.e.*, gradientForest) predicting genomic offset under future climate conditions. GradientForest implemented spatial modeling of genotype-climate relationship with two purposes: (1) linking genomic variants and climatic data to characterize how the allele frequencies of climate associated SNPs vary along the gradients of multiple climatic variables, and (2) projecting these genotype-climate relationships through time and calculate Euclidean distance between current and future genetic importance value (11, 28), which serves as a metric of genomic offset. We have added the methodological details to gradientForest analysis in the Methods (line 472-506) as below.

“We used gradientForest (28) to model compositional genetic turnover (*i.e.* turnover in allele frequencies) using nonlinear functions of climatic gradients. The turnover function transforms multidimensional climatic variables to multidimensional genetic space while selecting and weighting these variables such that they best summarize genomic variation (11, 28). We used only SNPs with minor allele frequency >0.05 because rare alleles tend to yield false positives (13). We collected 19 climate variables at 2.5-min resolution from WorldClim (64) (<http://www.worldclim.org>), and elevation from field records. We randomly selected 50,000 SNPs because of constraints in computational time, and for each SNP we used 500 regression trees to build a function for each of 20 climate variables. Only SNPs with $R^2 > 0$ (a measure of response of individual SNP to gradient of given climatic variable) were considered to be ‘predictive’ loci and were subsequently used in the aggregate turnover function, accounting for importance of climate variable and the goodness of fit for each SNP.

The gradient forest analysis provided a weighted R^2 value for each of 20 climatic variables to assess its importance (Table S4). We selected the top variables by ranking their importance and discarded those highly correlated (Pearson’s $r > 0.7$) with one variable with a higher weighted R^2 value. We then used turnover functions to examine change in allele frequencies along these variables and to transform them into genetic importance values. To visualize the resulting multidimensional genetic patterns in geographic and biological space, we used principal component analysis (PCA) to reduce the transformed climate variables into three principal components (11, 28).

We used the gradientForest outputs under current climate as the baselines to predict genomic variation under future climate conditions. The Euclidean distance between the current and predicted values is referred as genomic offset (11). To explore a range of potential future climate conditions, we used four climate models (CCSM4, MICRO-5, MPI-ESM-LR and CNRM-CM5-2) under four emission scenarios (RCP 4.5 and RPC 8.5 for 2050 and 2070 projections, Table S5) for a total of sixteen future climate conditions to predict genomic offset. To estimate the spatial regions where genotype–climate relationship will be most disrupted under future climate, we transformed the climate variables from each of the sixteen climate conditions into genetic importance using the turnover functions as described above. For each grid, we calculated the Euclidean distance between current and future genetic importance value (11, 28), which serves as a metric of genomic offset. We mapped the mean values from the sixteen future climate conditions to indicate the spatial distribution of genomic offset to climate change”.

Q3. I don’t like the terminology of adaptive genetic variation when all the authors have done is associate loci with environmental variables. Whether or not these loci are indeed adaptive will depend on how they influence fitness in the wild. The authors assume because there is a signature of selection these genes are adaptive and while I don’t disagree with this premise big chunks of these genes might not contribute to fitness. The only true way is to link SNP’s to fitness and hence phenotypes.

Response: The reviewer raised an issue of inappropriate use terminology of “adaptive genetic variation”. In our study, we carried out genotype–climate association and identified SNPs

correlated with climatic variables. The intraspecific variation in genotype-climate relationship has been used to predict genomic offsets under future climate conditions. In our previous version of manuscript, we have used the term of “adaptive genetic variation in climate adaptation” to refer this intraspecific variation in genotype-climate association.

As the reviewer has pointed out, genetic variants significantly associated with climate variables are not equal with “adaptive genetic variants”. We have realized that this inappropriate terminology (*i.e.*, adaptive genetic variants, adaptive potentials and fitness) has caused confusion in the previous version of manuscript. We have thus removed these inappropriate terms, *i.e.*, “adaptive genetic variants”, “adaptive potentials” and “fitness” from the revised version of the manuscript. In addition, we have changed “adaptive capacity” to “intraspecific variation in genotype-climate association”.

Minor comments:

Q4. Line 19 and 24 in rather than into

Response: We have changed (line 21 and 22).

Q5. Line 26 I don't understand what the authors mean by climate-adaptive populations.

Response: We have changed “climate-adaptive populations” to “the populations tolerant climate change”, as these populations have least genome-niche interruption under the future climate change (line 28-29).

Q6. Line 44 The authors should cite Bush et al Ecology Letters who take estimates of additive genetic variance and hence the evolutionary potential of phenotypes and integrate these estimates into species distributional models. As far as I am aware no one has been able to link genomic signatures of selection to the evolutionary potential of an organism, which is an organisms potential to respond to selection.

Response: Thank you for suggesting this interesting paper. Bush et al. (2016) developed a new generic modeling approach (AdaptR) that incorporates adaptive capacity through phenotypic plasticity and evolutionary adaptation into species distribution modeling, using 17 species of Australian fruit flies. This system considered evolutionary adaptation selectively for a species' critical thermal maximum (CTmax) and collected specific values for CTmax from previous studies (*i.e.*, Blackburn et al. 2014). The genetic variances for heat resistance were estimated through a full-sib–half-sib study design (10 species) as well as inferred from closely related species (the remaining 7 species, Kellermann et al. 2012). This study provides an ideal system to incorporate multiple factors in predicting climate-change driven vulnerability, in particular considering evolutionary potential.

With a great potential, such a well-designed system can only be carried out with well-studied model organisms, *e.g.*, fruit flies as in Bush et al., for which trait changes under different environmental conditions as well as genealogical data can be obtained. This may explain why

not many studies can link phenotypic change, fitness, selected genetic variants and organisms' evolutionary potentials in climate-change driven vulnerability estimates. We do agree with the reviewer that this system of incorporation multiple explanatory factors has a great potential for the field. We have thus highlighted these potential directions that are important for improving the climate-change driven vulnerability estimates, and cited Bush et al. 2016 (ref. 57) as an example in the Conclusion and Perspective (lines 414-422) as follows.

“In essence, the actual evolutionary responses of species to climate change are more complex than what explained by aforementioned three factors. It requires understanding interactions between local adaptation, **phenotypic plasticity, evolutionary potential**, the effective population size, dispersal ability and interspecific interactions (*i.e.*, 15, 48, **57**). Consequently, a combination multiple predictors will improve our understanding of climate-change driven species vulnerability. A further implement of physiological tolerance experiments of species, *i.e.*, common garden and transplant experiments (58), as well as a functional test of climate sensitive genes (46), would shed more light on how genetic adaptation actually leads to climate-adapted fitness”.

Q7. Line 46 This is not an accurate definition of adaptive capacity.

Response: We agree with the reviewer and have removed this paragraph of “adaptive capacity”. Instead, we focus on the intraspecific variation in genotype-climate association and how this can be translated into prediction of climate-change driven vulnerability (lines 50-56) as below.

“These ecological genomic studies have shown different genotype-climate associations between populations and suggested that local populations may respond differentially to climate change (11-13). Nevertheless, the intraspecific variation has rarely been incorporated into ecological niche models that often assume a uniformity of climate response between populations (14-16). There is a urgent need to incorporate intraspecific genomic variation in modeling habitat suitability in the context of climate change, as such information is necessary for understanding fine-scale estimate of climate-change driven vulnerability (*e.g.*, 17)”.

Q8. Line 51 But this also assumes that there is no gene flow between populations and assumes that populations do not possess the underlying genetic variation to adapt to climate change. Perhaps they have alleles, at low frequency because they are not under strong selection but as the environment changes these alleles will rise in frequency. If any populations are “better” adapted to future climates these alleles will be passed to the vulnerable populations.

Response: The reviewer raised an important issue if gene flow existed between different climate-tolerant groups thus climate-associated alleles can be exchanged. We think this is very important and needs to address carefully.

For the two studied species, we observed mixed genotypes between the three groups, suggesting the possibility of gene flow. As these admixed individuals are only observed in the

areas where the three genetic groups are in contact (Fig. S8), we suspect that the genetic admixture is likely to be the consequence of historical gene flow during glacial periods, when these groups expanded to the valleys at low elevation and got secondary contact (25-26). To explicitly test this possibility, we compared three demographic models (a zero gene flow model, a historical gene flow model and a continuous gene flow model, Fig. S9 and Supplementary note 2) using Fastsimcoal v2 (50). Our model test supports a historical gene flow model with migration (probability between lineages on a per generation basis, *T. elliotii*, $6.7e^{-6}$ - $3.07e^{-3}$; *P. monticolus*, $7.56e^{-5}$ - $1.35e^{-3}$) occurring between 11.5-15.5 thousand years ago (kya, *T. elliotii*, 11.5-16.8 kya; *P. monticolus*, 13.3-17.5kya, Tables S9-10). After the ice age, gene flow ceased when the three groups were pushed to higher elevations. Considering that these groups are currently isolated within different mountain systems and that they are likely to become pushed uphill when climate gets warmer, the climate-associated genotypes may not easily spread across distribution ranges and obscure the observed intraspecific genomic variation.

We have added this part of information in the Discussion (lines 335-350). We have also provided a detailed description of methods and results in a supplementary note (note 2), a figure (Fig. S9) and two tables (Tables S9-10).

Q9. Line 56 Revise sentence

Response: We have revised this description in lines 57-64 as follows.

“Mountainous areas harbor an exceptional biodiversity and endemism but are highly vulnerable to climate change (18). This is because that the complex topography within a rather small geographical area leads to dramatic ecological stratification in the mountains. Species are often restricted into spatially heterogeneous environments and locally adapted to diversified climate conditions (18-20). These population-specific adaptations likely drive different responses under climate change because the populations likely track their own optimally environmental conditions (20). Despite a great potential, this intraspecific variation has not been considered into vulnerability estimates driven by climate change (12-13)”.

Q10. Line 60 Have studies looked at the birds you study here and looked at their capacity to respond plastically or even how the traits vary across populations? Is their great potential for plasticity, I guess you could say that in lots of birds there is evidence for plasticity in traits X,Y and Z which we think are going to be important for climate change so we make the assumption that the same is for these species but it is still an assumption.

Response: The reviewer raised an important point that plasticity has a great potential for climate change. It is true that phenotype plasticity has contributed to response to climate change (*i.e.*, see Valladares et al. 2014; Foden et al. 2019). Unfortunately, as there is no any documented phenotypic plasticity in these two mountainous species, we are unable to consider this in our study.

However, we do agree with the reviewer that adding the dimension of phenotypic plasticity

could further improve predictions of climate-change driven vulnerability. We have thus highlighted these potential directions as important for further development in the Conclusions and Perspectives subsection (lines 414-422) as follows.

“In essence, the actual evolutionary responses of species to climate change are more complex than what explained by aforementioned three factors. It requires understanding interactions between local adaptation, **phenotypic plasticity**, evolutionary potential, the effective population size, dispersal ability and interspecific interactions (*i.e.*, 15, 48, 57). Consequently, a combination multiple predictors will improve our understanding of climate driven species vulnerability. A further implement of physiological tolerance experiments of species, *i.e.*, common garden and transplant experiments (*i.e.*, 58), as well as a functional test of climate sensitive genes (*i.e.*, 46), would shed more light on how genetic adaptation actually leads to climate-adapted fitness”.

Q11. Line 68 *But here you say you know nothing about their adaptive potential.*

Response: We agree with the reviewer and have changed this sentence to “There is an urgent need to refocus conservation efforts toward this threatened region, but such efforts are hampered by a lack of knowledge of ecological genomics and the niche suitability of mountainous species that underpin such strategies ” in lines 70-72.

Q12. Line 77 *Is there gene flow between these populations? There is a difference between local adaptation, which you can test by common garden or reciprocal transplant experiments to look at how fitness changes, and isolated populations that no longer mix.*

Response: The reviewer raised two issues in this comment and we respond them one by one.

- a) The first issue that the reviewer raised is whether there is gene flow between the three groups. As there are mixed genotypes only observed in the areas where the three genetic groups are in contact (Fig. S8), we suspect the genetic admixture to be the consequence of historical gene flow during glacial periods, when these groups expanded to the valleys at lower elevation and got into secondary contact (25-26). To explicitly test this, we compared three demographic models (a zero gene flow model, a historical gene flow model and a continuous gene flow model, Fig. S9 and Supplementary note 2) using Fastsimcoal v2 (50). The results support a historical gene flow model with migration (probability between two groups on a per generation basis, *T. elliotii*, $6.7e^{-6}$ - $3.07e^{-3}$; *P. monticolus*, $7.56e^{-5}$ - $1.35e^{-3}$) occurring between 11.5-15.5 thousand years ago (kya, *T. elliotii*, 11.5-16.8 kya; *P. monticolus*, 13.3-17.5kya, Tables S9-10). Considering that these genetic groups are currently isolated within different mountain systems and that they are likely to become pushed uphill when climate gets warmer, the climate-associated genotypes may not easily spread across distribution ranges and obscure the observed intraspecific genomic variation. We have added this part of analysis to the Discussion (lines 335-350), and also provided a detailed description of methods and results in a supplementary note 2, Fig. S9 and Tables S9-10.

- b) The second issue the reviewer asked is to test fitness changes and gene flow using common garden or reciprocal transplant experiments. While this is a great idea, it is currently logistically impossible because of several obstacles.

The first obstacle is to implement such experiments on wild birds. Although these experiments are commonplace in plants and a few model animals (*e.g.*, zebra fish, chicken and mice), these experiments are very difficult and time-consuming to undertake on wild species and it is highly unsure they would deliver results in a sufficiently timely manner (Rellstab et al. 2015). Even if we could overcome the logistical challenges of conducting long-term experiments on the Southwest Mountains, it is very difficult to raise and breed wild birds in captivity.

Apart from the logistical challenges, there are still the obstacles to find suitable traits related to fitness in wild species. Phenotypic changes observed in common garden and reciprocal transplant experiments can be driven by plasticity, thus a direct inference of which phenotypic change is related to local adaptation is less likely without laborious and often unfeasible experiments (Rellstab et al. 2015). In addition, fitness is not easy to measure because fitness differs from trait to trait, and from species to species, and it is also bound to the experimental conditions (site conditions, duration and age of individuals). Many traits are the result of polygenic adaptation that also puts additional challenge for testing fitness of genetic variants.

Our aim in this study is to develop an integrative framework to facilitate wildlife conservation under increasingly global warming. This approach should be strategically easy to transfer to other species. Unfortunately, set up experimental designs of common garden and transplant experiments to test fitness is not strategically possible for the wild animals. Nevertheless, we do agree with the reviewer that an implementation of common garden and transplant experiments in certain focal species could provide direct evidence for estimating climate-adapted fitness. We have thus added this potential dimension for the future development in Discussion and Perspectives (lines 414-422) as below.

“..., the actual evolutionary responses of species to climate change are more complex than what explained by aforementioned three factors. It requires understanding interactions between local adaptation, phenotypic plasticity, evolutionary potential, the effective population size, dispersal ability and interspecific interactions (*i.e.*, 15, 48, 57). Consequently, a combination multiple predictors will improve our understanding of climate-change driven species vulnerability. **A further implement of physiological tolerance experiments of species, *i.e.*, common garden and transplant experiments** (*i.e.*, 58), as well as a functional test of climate sensitive genes (*i.e.*, 46), would shed more light on how genetic adaptation actually leads to climate-adapted fitness”.

Q13. Line 122 Is this fairly standard number of individuals for estimating population-level variation. I noticed some populations are captured by only 2 individuals which seem low for capturing population-level variation (Table S1). Is there a link between genomic

vulnerability and the number of individuals sampled? It would be good to have warm and cold-adapted environments put into Table S1.

Response: Thank you for pointing out the issue of the limited sample size in a few localities. Ideally one should have at least four individuals from each locality in the gradientForest analysis (e.g., Ruegg et al. 2018). Considering this, we have worked hard to find more samples from other museums, and have managed to increase our sampling to 55 and 58 individuals for *T. elliotii* and *P. monticolus*, respectively. However, the studied species are fairly rare in the highly heterogeneous mountainous areas. The samples used in this study involve a great effort over many decades. Even though we have obtained a good sampling for *P. monticolus*, we could only collect two or three individuals of *T. elliotii* for three of the localities (i.e., Rantang, Mangkang and Wolong).

We decided to keep these localities despite the low sample sizes because they are situated in the adjacent areas of the three genetic groups, which would help to clarify the division line and the degree of the genetic admixture of the groups. Given this, we implemented a deep-sequencing strategy because low precision in statistical inference due to the low sample size can be offset by using a large number of SNPs in the analyses (65-66). In addition, we have carried out a parallel genomic offset analysis using Generalized Dissimilarity Modeling (GDM, Ferrier et al. 2007), a distance-based method (i.e., F_{ST}) to estimate relationship between genomic variation and climatic variables because F_{ST} analysis provides reliable estimates even for populations with small sample sizes (i.e., two individuals) given a large number of SNPs (i.e., >1500) (66, 68). Our GDM analyses show similar genomic offset patterns to those of gradientForest analyses for the two species, suggesting that our genomic offset results are robust to low sample sizes at a few localities. We have added this part of text to Methods (lines 508-527) and provided a detailed description in the Supplementary note 3 and Fig. S10. We have also provided a column in Table S1 to define the cold-dry, warm-humid and warm-dry tolerant groups.

Q14, Line 136 By genomic variations you mean SNP?

Response: Yes, these genomic variants are SNPs. We realize that “genomic variation” is actually “genomic variants” and have changed (line 130).

Q15. Line 143 I don't see any environmental variable related to precipitation as being amongst your 3 most important environmental variables.

Response: Thank you for pointing out this missing information. After increasing sampling, we have re-analyzed gradientForest. In the revised version of manuscript we have updated the top climatic variables that are associated with genomic variants in lines 130-138 as below.

“Of the 20 climatic variables tested (Table S3), the top five uncorrelated explanatory variables identified were BIO3 (isothermality, i.e., mean diurnal range divided by the temperature annual range), BIO18 (precipitation of the warmest quarter), BIO9 (mean temperature of the driest quarter), BIO19 (precipitation of the coldest quarter) and BIO5 (max temperature of the warmest month) for *P. monticolus*, while the top six climatic variables for *T. elliotii* were

BIO2 (mean diurnal range, *i.e.*, the mean of the monthly differences between the maximum and minimum temperatures), elevation, BIO1 (annual mean temperature), BIO7 (temperature annual range), BIO17 (precipitation of the driest quarter) and BIO4 (temperature seasonality) (Fig. 3a-b and Table S4)".

Q16. Line 144 *I don't understand the authors logic here, are they saying that because the environment was found to associate with SNP's these species are more sensitive to climate change? I think most organisms associate with the environment in one way or another.*

Response: Species in which the majority of populations already live under conditions that are close to their physiological thresholds are likely to be at higher risk from climate change (Foden et al. 2007). As species in the mountains and the desert face a large temperature and precipitation amplitude year round, they are likely already close to the thresholds beyond which physiological functions quickly break down, *i.e.*, drought-tolerant desert plants (Foden et al., 2007) and high temperature-tolerant birds (Cunningham et al., 2013).

We have realized that this inference is a bit far-reaching considered our results, therefore we have removed this inference from the revised version of manuscript.

Q17. Line 149 *It is really difficult for me to understand the geography of your study populations and all I want to know is there gene flow between these populations?*

Response: Thank you for pointing out the unclear geographical definition in study areas. We have now delineated ecological zones to facilitate the understanding of genotype-climate association (*i.e.*, Fig. 2 and 3). We outlined these ecological zones because mountains areas are topologically complex and spatially heterogeneous. Even in small geographic areas, these species show different genotype-climate associations across various ecological zones. This intraspecific variation was further used to estimate genomic offsets.

The reviewer also asked about gene flow among the genetic groups. As we observed mixed genotypes in the areas where the three genetic groups are in contact (Fig. S8), we suspect that the genetic admixture is likely to be the consequence of historical gene flow during glacial periods, when these groups expanded to the valleys at low elevation and got into secondary contact (25-26). In order to test this, we have compared three demographic models using Fastsimcoal v2 (50), a zero gene flow model, a historical gene flow model and a continuous gene flow model (Fig. S9 and Supplementary note 2). Our model test supports a historical gene flow model with migration (probability between lineages on a per generation basis, *T. elliotii*, $6.7e^{-6}$ - $3.07e^{-3}$; *P. monticolus*, $7.56e^{-5}$ - $1.35e^{-3}$) occurring between 11.5-15.5 thousand years ago (kya, *T. elliotii*, 11.5-16.8 kya; *P. monticolus*, 13.3-17.5kya, Tables S9-10). After the ice age, gene flow ceased when the three groups were pushed to higher elevations. Considering that these genetic groups are currently isolated within different mountain systems and that they are likely to become pushed uphill when climate gets warmer, the climate-associated genotypes may not easily spread across distribution ranges and obscure the observed intraspecific genomic variation.

We have added this part of analysis in the Discussion (lines 335-350).

Q18. Line 153 *What do you mean here, did different environmental variables associate with species from the northern edges?*

Response: This sentence has been rephrased to “We found that the genotype-climate associations vary from the western parts (*i.e.*, the *Southern-Tibetan zone*, *East-Himalayan zone*) to the eastern (*i.e.*, the *Western mountainous plateau zone*) and southern parts of the distribution ranges (*i.e.*, the *Southwest mountainous zone*) (Fig. 3a-b)” in lines 145-148.

Q19. Line 154 *I don't think you can say this, who know what genes will underpin climate change responses.*

Response: We have rephrased this sentence to “These results suggest that the two species show intraspecific variation in their genotype-climate associations, which are likely subject to local adaptation to heterogeneous climatic conditions” in Results (lines 148-150).

Q20. Line 161 *I don't understand how the authors are predicting future genomic compositions and their methods didn't help me understand this either.*

Response: Thank you for pointing out this lack of information about the gradientForest method, which we now have provided a detailed description (lines 472-506) as follows.

“We used gradientForest (28) to model compositional genetic turnover (*i.e.* turnover in allele frequencies) using nonlinear functions of climatic gradients. The turnover function transforms multidimensional climatic variables to multidimensional genetic space while selecting and weighting these variables such that they best summarize genomic variation (11, 28). We used only SNPs with minor allele frequency >0.05 because rare alleles tend to yield false positives (13). We collected 19 climate variables at 2.5-min resolution from WorldClim (64) (<http://www.worldclim.org>), and elevation from field records. We randomly selected 50,000 SNPs because of constraints in computational time, and for each SNP we used 500 regression trees to build a function for each of 20 climate variables. Only SNPs with $R^2 > 0$ (a measure of response of individual SNP to climatic gradient of given variable) were considered to be ‘predictive’ loci and were further used in the aggregate turnover function, accounting for importance of climate variable and the goodness of fit for each SNP.

The gradient forest analysis provided a weighted R^2 value for each of 20 climatic variables to assess its importance (Table S4). We selected top variables by ranking their importance and discarded those highly correlated (Pearson's $r > 0.7$) with one variable with higher weighted R^2 value. We then used turnover functions to examine change in allele frequencies along these variables and to transform them into genetic importance values. To visualize the resulting multidimensional genetic patterns in geographic and biological space, we used principal component analysis (PCA) to reduce the transformed climate variables into three principal

components (11, 28).

We used the gradientForest outputs under current climate as the baselines to predict genomic variation under future climate conditions. The Euclidean distance between the current and predicted values is referred as genomic offset (11). To explore a range of potential future climate conditions, we used four climate models (CCSM4, MICRO-5, MPI-ESM-LR and CNRM-CM5-2) under four emission scenarios (RCP 4.5 and RPC 8.5 for 2050 and 2070 projections, Table S5) for a total of sixteen future climate conditions to predict genomic offset. To estimate the spatial regions where genotype-climate relationship will be most disrupted under future climate, we transformed the climate variables from each of the sixteen climate conditions into genetic importance using the turnover functions as described above. For each grid, we calculated the Euclidean distance between current and future genetic importance value (11, 28), which serves as a metric of genetic offset. We mapped the mean values from the sixteen future climate conditions to indicate the spatial distribution of genomic offsets to climate change”.

Q21. Line 175 Are the authors using the same measure as Fitzpatrick and Keller: genomic offset?

Response: Yes, “genomic vulnerability” used in the previous version of manuscript is “genomic offset” in Fitzpatrick and Keller (2015). We have now changed to “genome offset” throughout the revised version of manuscript.

Q22. Line 183 Is this just based on the fact that the environment is expected to shift under climate change to a greater degree than other environments? I am failing to see what the genomic angle brings to the table because if that is the case then simply modelling the expected change in environment would tell you the same thing.

Response: The reviewer raised an interesting question about whether the different genomic offsets observed in the three climate tolerant groups are expected to be the consequence of different degrees of climate changes in the environments.

In order to address this issue, we compared the climate change between niches where harboring the cold-dry, warm-humid and warm-dry tolerant populations using the climatic variables identified in the gradientForest analysis. We calculated the difference between future and current climatic variable values and then compared those values between the three groups. A positive value shows an increase and a negative value shows a decrease in variable change. Our comparisons show complex patterns of climate change under future climate conditions for the two species (Figure 1), as the changes differ from variables to variables and from climatic scenarios to climatic scenario (2050RCP4.5, 2050RCP8.5, 2070RCP4.5 and 2070 RCP8.5). Thus, as future climate change seems a complicate process and causes different effects on these climatic variables, it is not straightforward to determine which regions or which variables show larger degrees of climate change.

As the gradientForest and GDM analyses show different genotype-climate associations and genomic offsets between the three climate tolerant groups, a more parsimonious explanation would be that these populations locally adapt to different climatic conditions and thus have different genomic offsets under future climate change.

T.elliotii

P. monticolus

Figure 1. Differences between climatic variables under current and future climate among the three climate-tolerant groups. The ranges of these values are scaled by dividing by one standard deviation.

Q23. Line 195 Do these genetic clusters match up with the environmental conditions? Could you not just say this by looking at the species distributions and environments they are found in?

Response: The reviewer asked the important question about whether the identified genetic groups match the environmental conditions where they live. The distributions of the three groups are generally congruent with the ecological zones described in the Methods (Study areas). For example, cold-dry tolerant group of *T. elliotii* is in the *Southern-Tibetan zone*, which is located in the southeastern part of the Qinghai-Tibetan Plateau that has an average elevation of 4,500 m a.s.l. (STZ in Fig.2a). This zone is dominated by a plateau with cold-dry climate and typical alpine meadow and shrub land. The warm-dry populations are mostly in the *Western mountainous plateau zone* (WMPZ in Fig.2a), where is in the northeast range of the Southwestern Mountains and is mostly comprised of mid-elevational mountains with typical temperate climate.

Nevertheless, it is not always a close match between genetic groups and the eco-regions. The *Southwest mountainous zone* (SMZ in Fig.2a) is highly heterogeneous and has a vertical climatic zonation ranging from subtropical broadleaf forests, temperate coniferous forests to the alpine meadow. The populations of *T. elliotii* in this zone show quite different genotype-climate associations and genomic offsets under future climate conditions, as those

in the temperate coniferous forests are clustered into cold-dry tolerant group and those in the tropical broadleaf forests are clustered into warm-humid tolerant group (Fig. S7). Considering highly heterogeneous mountain environments, a detailed investigation of intraspecific variation in genotype-climate association is necessary to unravel potentially climatic variables and genotypes in order to predict climate-change driven genomic offsets. We have added this part of text in lines 317-333.

Q24. Line 205 Is this not circular? You used genomics to define the groups and then use the groups to define genomic vulnerability?

Response: The issue whether the defining groups and genomic offset estimate become circular in argument is important. To define climate tolerant groups we used LFMM, RAD and dbRAD to detect the SNPs that are significantly related to the top climatic variables. Based on these SNPs we carried out a population genetic structure analysis using Admixture to cluster populations into different groups.

As the population genetic structure analyses identified three clusters in each species, we want to know how these groups (clusters) respond to future climate conditions, for example, which group has large genomic offset and niche suitability decline, which group has minor genome offset and niche suitability interruption, and, if the genomic offset and niche suitability change show congruent patterns. In brief, the population genetic structure analysis is a pre-requirement to find the different groups, and genomic offset and niche suitability change comparisons are aimed to find the groups with minor climate change interruption. We thus think that defining groups and estimating their respective genomic offsets are not circular in argument.

Q25. Line 211 I don't quite understand the author's argument here around intraspecific variation.

Response: Here the intraspecific variation referred to three climate tolerant groups identified by population genetic structure analysis. We have clarified this (lines 226-227) as below.

“We next investigated whether the predicted changes in niche suitability caused by the future climate conditions differ among the three climate tolerant groups”.

Q26. Line 244 I am still wondering what is the gene flow between these populations because surely if there is gene flow then populations with low genomic vulnerability will simply mate with populations with high genomic vulnerability?

Response: The reviewer asked whether there is gene flow between the three genetic groups allowing climate-associated alleles to spread between populations. As we address in R3Q8 and R3Q17, we have used Fastsimcoal v2 to test three demographic models, a null model without gene flow, a model of continuous gene flow and a model of historical gene flow. In the continuous gene flow model, three groups were assumed to exchange gene flow freely, thus satisfied the hypothesis of which climate-associated alleles can spread between

populations. We have also set up a model of historical gene flow because we only observed genetically admixed individuals in where the three groups are in contact (Fig. S8).

Our model test supported a model of historical gene flow with migration occurring between 11.5-15.5 thousand years ago (Tables S9-10). This gene flow likely occurred during glacial periods when these groups dispersed into lower elevations and got in contact (25-26). After the ice retreated, the three groups were pushed to higher elevation and got isolated again. Based on these results, we considered that the climate-associated genotypes cannot spread freely between the groups, particularly given that future climate warming most likely will push populations uphill and further facilitate the geographic isolation between populations. We have added this part of analysis to Discussion (lines 335-350). We have also provided a detailed description about this part of analysis in Supplementary note 2, Fig. S9 and Tables S9-10.

Q27. Line 263 Should a discussion of this go into the intro? I felt like the author's results was a results/discussion and a lot of their discussion was a re-hash of their results

Response: Thank you for suggestion and we have removed this sentence to Introduction. The reviewer also pointed out that some parts of the results are repeated in the Discussion. Our study is comprised of three parts, *i.e.*, genomic offset, niche suitability and landscape genetics. In the Discussion, we wanted to bring all these results in a concerted way to show how and why these results are congruent or incongruent. This ambition may have brought some unnecessary details into the manuscript and we have now streamlined the Discussion and removed the some repeated results.

Q28. Line 278 Precipitation was not listed as important above? And if precipitation is important is it worth noting that our understanding of how precipitation will change in the future is less clear.

Response: As the reviewer pointed out, some results were repeated in Discussion. We have removed the description of these variables in Discussion and instead presented a full list of the top climatic variables identified in the gradientForest in Results (lines 130-138).

Q29. Line 281 I don't think the authors are testing theories around niche conservatism. They also have not measured thermal tolerance but are just assuming that their correlations between environment and genes translate into thermal tolerance and fitness differences.

Response: We agree with the reviewer and have removed this part of text in the revised version of manuscript.

Q30. Link 333 Did you need genomics to get to this conclusion?

Response: We used landscape genetic approach to infer how landscape features would affect

the patterns of gene flow between populations, that is, what are the potential dispersal routes between populations. Our landscape genetic analysis used genetic distance as the dependent variable and resistance costs from the landscape features as independent variables. To obtain a matrix of genetic distance we randomly extracted 100,000 SNPs from inter-genic regions of the genomes to calculate genetic distance (F_{ST}) between populations. We explain this in lines 659-660.

References:

- Araújo, M. B., & New, M. (2007). Ensemble forecasting of species distributions. *Trends in Ecology & Evolution*, *22*, 42–47.
- Araújo, M. B., Anderson, R. P., Barbosa, A. M., Beale C. M., Dormann, C. F., Early, R., ..., Rahbek, C. (2019). Standards for distribution models in biodiversity assessments. *Science Advances*, *5*, eaat4858.
- Bay, R. A., Harrigan, R. J., Underwood, V. L., Gibbs, H. L., Smith, T. B., & Rugg, K. (2018). Genomic signals of selection predict climate-driven population declines in a migratory bird. *Science*, *359*, 83–86.
- Blackburn, S., van Heerwaarden, B., Kellermann, V. & Sgrò C. M. (2014). Evolutionary capacity of upper thermal limits: beyond single trait assessments. *Journal of Experimental Biology*, *217*, 1918–1924.
- Bush, A., Mokany, K., Catullo, R., Hoffmann, A., Kellermann, V., Sgrò C., McEvey, S. & Ferrier, S. (2016). Incorporating evolutionary adaptation in species distribution modeling reduces projected vulnerability to climate change. *Ecology Letters*, *19*, 1468–148.
- Cao, Y. H., Zhu, S. S., Chen, J., Comes, H. P., Wang, I. J., Chen, L. Y., Sakaguchi, S. & Qiu, Y. Y. 2020. Genomic insights into historical population dynamics, local adaptation, and climate change vulnerability of the East Asia Tertiary relict *Euptelea* (Eupteleaceae). *Evolutionary Applications*, *13*, 2038–2055.
- Chen, Y., Liu, Z., Régnière, J., Vasseur, L., Lin, J., Huang, S., ..., You, S. (2021). Large-scale genome-wide reveals climate adaptive variability in a cosmopolitan pest. *Nature Communications*, *12*, 7206.
- Fitzpatrick, M. C. & Keller, S. R. (2015). Ecological genomics meets community-level modelling of biodiversity: mapping the genomic landscape of current and future environmental adaptation. *Ecology Letters*, *18*, 1–16.
- Foden, W. B., Young, B. E., Akcakaya, H. R., Garcia, R. A., Hoffmann, A. A., Stein, B. A., ..., Huntley, B. (2019). Climate change vulnerability assessment of species. *WIREs Climate Change*, *10*, e551.
- Gotelli, J. N. & Stanton-Geddes, J. (2015). Climate change, genetic markers and species distribution modeling. *Journal of Biogeography*, *42*, 1577–1585.
- Kellermann, V., Overgaard, J., Hoffmann, A.A., Flojgaard, C., Svenning, J.C. & Loeschcke, V. (2012). Upper thermal limits of *Drosophila* are linked to species distributions and strongly constrained phylogenetically. *Proceedings of the National Academy of Sciences of the United States of America*, *109*, 16228–16233.
- Razgour, O., Forester, B., Taggart, J. B., Bekaert, M., Juste, J., Ibanez, C., ..., Manel, S. (2019). Considering adaptive genetic variation in climate change vulnerability assessment reduces species range loss projections. *Proceedings of the National Academy*

- of Sciences of the United States of America*, 116, 10418–10423.
- Rellstab, C., Gugerli, F., Eckert, J. A., Hancock, M. A., Holderegger, R. (2015). A practical guide to environmental association analysis in landscape genomics. *Molecular Ecology*, 24, 4348–4370.
- Rhoné B., Defrance, D., Berthouly-Salazar, C., Mariac, C., Cubry, P., Couderc, M., ..., Vigouroux, Y. (2020). Pearl millet genomic vulnerability to climate change in West Africa highlights the need for regional collaboration. *Nature Communications*, 11, 5274.
- Rose NH, Bay RA, Morikawa MK, Palumbi SR. 2018. Polygenic evolution drives species divergence and climate adaptation in corals. *Evolution*. 72, 82–94.
- Ruegg, K., Bay, R. A., Anderson, E. C., Saracco, J. F., Harrigan, R. J., Whitfield, M., ..., Smith, T. B. (2018). Ecological genomics predicts climate vulnerability in an endangered southwestern songbird. *Ecology Letters*, 21, 1085–1096.
- Smith, T. B., Fuller, T. L., Zhen, Y., Zaunbrecher, V. Z., Thomassen, H. A., Njabo, K., Anthony, N. M. & Gonder, M. K. 2021. Genomic vulnerability and socio-economic threats under climate change in an African rainforest bird. *Evolutionary Applications*. 14, 1239–1247.
- Valladares, F., Matesanz, S., Guihaumon, F., Ara újo, M. B., Benito-Garzón, M., Cornwell, W., ..., Zavala, M. A. (2014) The effects of phenotypic plasticity and local adaptation on forecasts of species range shifts under climate change. *Ecology Letters*, 17, 1351–1364.
- Van Strien, M. J., & Keller, D., Holderegger, R. (2012). A new analytical approach to landscape genetic modelling: least-cost transect analysis and linear mixed models. *Molecular Ecology*, 21, 4010–4023.
- Wiens, J. J. (2016). Climate-related local extinctions are already widespread among plant and animal species. *PLoS Biology*, 14, e2001104.
- Wood, G., arzinelli, E. M., Campbell, A. H., Steinberg, P. D., Verges, A., Coleman M. A. 2021. Genomic vulnerability of a dominant seaweed points to future-proofing pathways for Australia's underwater forest. *Global Change Biology*, 27, 2200–2212.

Reviewers' comments:

Reviewer #1 (Remarks to the Author):

The authors have adequately addressed my major concerns with their revised analyses, apart from the issue of sampling bias and spatial autocorrelation among location records used in the species distribution modelling. Is thinning records to 10 km threshold sufficient to reduce sampling bias and spatial autocorrelation across such a large study area? How has this been tested?

Reviewer #2 (Remarks to the Author):

I commend the authors for their careful and thorough attention to the concerns raised during the previous round of reviews. While I feel the study is improved and aspects of the Methods are now clear, some issues remain for consideration.

I am concerned about the use of elevation in the GF models and how it may have impacted genomic offset predictions. As Figure 3A shows, elevation ('alt') is an important correlate of genomic variation in *T. elliotii*. However, organisms don't respond to elevation per se, but rather to changes in climate associated with changes in elevation. When used to estimate genomic offset, elevation will not contribute at all because elevation remains constant (so the distance between the current and future transformed genomic spaces will be zero for this variable). As a result, genetic offsets for *T. elliotii* are possibly lower than what would be expected from a model fit with only climate variables.

It is unclear to me if the genomic offset models were fitted using outlier SNPs or simply those that had a positive R^2 from GF. Can this be clarified? Failure to correct for neutral population structure could be highly problematic.

L63-64: Authors state that "intraspecific variation has not been considered into vulnerability estimates driven by climate change", yet numerous papers have indeed done just this and the authors cite some of these papers in previous paragraphs. Please clarify.

L232-233: No clear what is meant by “...niche models have greater support and discrimination ability”. TSS and AUC are metrics of discrimination only, not support.

L236, Fig. 3c,d & elsewhere: It seems that the authors combined projections from multiple climate models AND emission scenarios AND decades. While it is recommended to use multiple climate forecasts, it is not defensible to combine forecasts across different emission scenarios and especially not across different decades. What is the justification for doing so?

L335-350: This paragraph takes up a lot of space in the MS but it is not clear how this is relevant to the broader study & comes across as a bit of a side story.

L408-410: I don't follow this statement. It is not known how either approach relates to climate change vulnerability, if at all. I would reword to state that by combining these methods, we can achieve unique insights that are not obtainable by either method in isolation. Note that this comment also calls into question the title of the paper, which has the same problem. There is no test of how well these methods work to predict vulnerability so one cannot claim if they are adequate or not.

L498-500: To be clear, genetic offsets were calculated for: four climate models, two emission scenarios (4.5 and 8.5) and two future decades (2050 and 2070). See comment above that it is not best practices to combine across emission scenarios and decades – these should be reported separately.

Fig. 2b – It is hard to see the different ranges and their overlap.

Reviewer #1 (Remarks to the Author):

Q1. The authors have adequately addressed my major concerns with their revised analyses, apart from the issue of sampling bias and spatial autocorrelation among location records used in the species distribution modelling. Is thinning records to 10 km threshold sufficient to reduce sampling bias and spatial autocorrelation across such a large study area? How has this been tested?

Response: The reviewer raised the question of whether a 10-km threshold is sufficient to reduce sampling bias and spatial autocorrelation across the study area. In the ecological niche modelling (ENM), we have tested how reducing sampling bias affected autocorrelation using a range of thresholds to thin distribution records, *i.e.*, from 5-km to 35-km with 5-km intervals. We used Moran's I to estimate the spatial autocorrelation in each thinning threshold. Compared to that of non-thinned records, we found that Moran's I decreased and reached a stable level at the approximately 10-km and 20-km thresholds (Figure 1 below). We then carried out two separate ENMs using records thinning with 10-km and 20-km thresholds, respectively, and we found that both analyses produced similar results (Figure 2 and Table 1 below).

We decided to present results based on the 10-km thinning threshold because of the high environmental heterogeneity of mountainous regions (following the recommendation of Borai et al. 2014). A 10-km resolution was considered to essentially limit the loss of environmental variation (by avoiding a too high thinning resolution, Anderson, 2012), but also to address sampling bias, Figure 1 below). This resolution has been used in previous ENMs in mountainous areas (*e.g.*, Pearson et al. 2007; Anderson and Raza 2010; Boria et al. 2014). To make it clear, we have added the relevant explanations to the Methods (lines 548-556), and provided a supplementary figure showing a plot of Moran's I against the different thinning thresholds (Fig. S10 in the revised manuscript).

Figure 1. Sampling bias and autocorrelation indicated by Moran's I. Moran's I decreased and reached to a stable level at the approximately 10-km and 20-km thresholds.

Figure 2. Projections of changes in niche suitability under future climate conditions using two separate ENMs based on the 10-km and 20-km thinning thresholds, respectively. Both analyses show similar results. The reddish colours show areas with increasing niche suitability (>0) and blue colours show areas with decreasing niche suitability (<0). The warm-humid and warm-dry tolerant groups are expected to experience a more severe future decline of the niche suitability than the cold-dry tolerant groups. Wilcoxon tests, ***, $P < 0.001$. Left, *T. elliotii*; right, *P. monticolus*.

Table 1. Model performance of the ENMs using 10-km and 20-km thresholds.

Species	Groups	AUC (10-km)	AUC (20-km)	TSS (10-km)	TSS (20-km)
T. elliotii	Cold-dry	0.928	0.915	0.75	0.742
	Warm-dry	0.961	0.967	0.878	0.894
	Warm-humid	0.964	0.938	0.859	0.795
P. monticolus	Cold-dry	0.938	0.934	0.814	0.826
	Warm-dry	0.905	0.902	0.704	0.691
	Warm-humid	0.904	0.888	0.71	0.696

Reviewer #2 (Remarks to the Author):

Q1. I commend the authors for their careful and thorough attention to the concerns raised during the previous round of reviews. While I feel the study is improved and aspects of the Methods are now clear, some issues remain for consideration. I am concerned about the use of elevation in the GF models and how it may have impacted genomic offset predictions. As Figure 3A shows, elevation ('alt') is an important correlate of genomic variation in *T. elliotii*. However, organisms don't respond to elevation per se, but rather to changes in climate associated with changes in elevation. When used to estimate genomic offset, elevation will not contribute at all because elevation remains constant (so the

distance between the current and future transformed genomic spaces will be zero for this variable). As a result, genetic offsets for *T. elliotii* are possibly lower than what would be expected from a model fit with only climate variables.

Response: The reviewer raised an issue of how the inclusion of elevation in the gradientForest analysis (GF) may have impacted the predictions of genomic offset. The reason we included elevation in the GF analysis is that elevation has been considered to be an explanatory variable in several previous GF analyses with a similar scope as our study (e.g., Fitzpatrick & Keller, 2015; Bay et al. 2018; Ruegg et al. 2018). We thus included elevation to detect positive SNPs in the genotype environment association both because the three lineages of the two species differ in their elevational distribution, and in response to a question raised by the reviewer 3 in the first round review concerning local adaptation. We certainly agree with the reviewer that elevation in itself cannot contribute to future climate change. Considering this we had also carried out a generalized dissimilarity modeling (GDM) estimating genomic offsets, in which elevation was not included (Fig. S4 in the previous version). The two genomic offset analyses produced similar results regardless if elevation was included or not, suggesting that the main conclusion from the genomic offset comparisons are robust.

Concerning the GF analysis, the reviewer asked whether the estimated genomic offsets differ between the model fit with only climatic variables and the one with both climatic variables and elevation. As the two models identify different sets of top environmental variables (this was the case for *T. elliotii*) and identify different numbers of R^2 positive SNPs (the case for both *T. elliotii* and *P. monticolus*, Table 2 below), it is not possible to make a direct comparison of the genomic offsets from the two models. Instead, we have compared the genomic offsets among the three climate-tolerant groups in each model and found that the two models produce similar results, suggesting that our main conclusion is robust (Figure 3 below). To avoid confusion about the role of elevation in the analysis, we have replaced the GF results by the model fit with only the 19 climatic variables (e.g., lines 127-135, Fig. 3, Fig. 4, Fig.S4 and Table S4 in this revision).

Table 2. The top environmental variables and the numbers of R^2 positive SNPs identified by the model fit with 19 climatic variables and the one with 19 climatic variables and elevation. Note that the weighted R^2 value and number of R^2 positive SNPs can be different even for the same variables identified by both models because the numbers of input variables have been changed (i.e., 19 vs. 20 environmental variables).

Species	GradientForest models	Selected top variables	R^2 positive SNP number
T. elliotii	19 climatic variables	BIO2, BIO10, BIO7, BIO19, BIO4	5446
	19 climatic variable + elevation	BIO2, elevation, BIO1, BIO7, BIO17, BIO4	5476
P. monticolus	19 climatic variables	BIO3, BIO18, BIO09, BIO19, BIO5	7294
	19 climatic variable + elevation	BIO3, BIO18, BIO09, BIO19, BIO5	7354

Figure 3. The two gradientForest models produced similar results regardless if elevation was included (right) or not (left), suggesting that the gradientForest analysis results are robust. In both models, the cold-dry tolerant groups show significantly greater genomic offsets than the warm-dry and warm-humid tolerant groups. Wilcoxon test, ***, $P < 0.001$.

Q2. It is unclear to me if the genomic offset models were fitted using outlier SNPs or simply those that had a positive R^2 from GF. Can this be clarified? Failure to correct for neutral population structure could be highly problematic.

Response: We have used both R^2 positive SNPs and outlier SNPs to carry out the gradientForest analyses, and have clarified this in the lines 209-215 (Results). We have provided figures showing their respective results (R^2 positive SNPs, Fig. 4g-h and Fig. S4a; outlier SNPs, Fig. 4j-l and Fig. S4b). These details have also been clarified in the figure legends.

In addition, we have run two generalized dissimilarity modeling analyses to estimate genomic offsets using both the R^2 positive SNPs (Fig. S5a) and outlier SNPs (Fig. S5b). These results have been added to lines 215-221 (Results) and Fig. S5.

Q3. L63-64: Authors state that “intraspecific variation has not been considered into vulnerability estimates driven by climate change”, yet numerous papers have indeed done just this and the authors cite some of these papers in previous paragraphs. Please clarify.

Response: We have changed this sentence to “intraspecific variation has not been considered into vulnerability estimate driven by climate changes in the mountainous species” in lines 62-64. We have also removed the two citations.

Q4. L232-233: No clear what is meant by “...niche models have greater support and discrimination ability”. TSS and AUC are metrics of discrimination only, not support.

Response: We have removed “support”.

Q5, L236, Fig. 3c,d & elsewhere: It seems that the authors combined projections from multiple climate models AND emission scenarios AND decades. While it is recommended to use multiple climate forecasts, it is not defensible to combine forecasts across different emission scenarios and especially not across different decades. What is the justification for doing so?

Response: Thanking you for pointing out this issue. It is true that different emission scenarios and decades shouldn't be combined. In our previous version of manuscript, we had produced a total of 16 genomic offset maps for each species under different climate models, emission scenarios and decades, and all these maps showed similar results. We then decided to present a figure with a combination of the results in the main figures (*i.e.*, Fig. 3c-d in previous version) and place the results of the separate analyses in supplementary materials (*i.e.*, Fig. S1 in the previous version).

In the revised manuscript, we have presented the predictions from the different emission scenarios and decades independently. We have shown the projection under RCP8.5 2050

(the one having intermediate genomic offsets across the different emission scenarios and decades) as an example in the main figure (Fig. 3c-d, Fig. 4g-j and Fig. 5a-d in the revision), and provided other projections under the RCP4.5 2050, RCP4.5 2070 and RCP8.5 2070 as supplementary figures (*i.e.*, Fig. S1, S4, S5, S6 and S7 in the revision).

Q6. L335-350: This paragraph takes up a lot of space in the MS but it is not clear how this is relevant to the broader study & comes across as a bit of a side story.

Response: This paragraph addressed whether gene flow would impact the intraspecific variation among the three climate-tolerant groups. In the first round of review, the reviewer 3 and the editor both raised this issue. We have thus carried out a demographic analysis and discussed the possibility. As the reviewer pointed out, we realize that the context where this paragraph was placed disturbs the logic flow, and we have instead placed it in supplementary Note 3.

Q7. L408-410: I don't follow this statement. It is not known how either approach relates to climate change vulnerability, if at all. I would reword to state that by combining these methods, we can achieve unique insights that are not obtainable by either method in isolation. Note that this comment also calls into question the title of the paper, which has the same problem. There is no test of how well these methods work to predict vulnerability so one cannot claim if they are adequate or not.

Response: Thank you for suggesting this! A point of “a combination of genomic offset and ecological niche modeling providing unique insights in climate change-driven vulnerability” is a good description of our findings and we have changed the texts to “we demonstrate that a combination of genomic offset and niche suitability provides an unique insight into climate change-driven vulnerability estimates” in lines 388-389.

We have also changed the title to “The combination of genomic offset and niche modeling provides insights into climate change-driven vulnerability”.

Q8. L498-500: To be clear, genetic offsets were calculated for: four climate models, two emission scenarios (4.5 and 8.5) and two future decades (2050 and 2070). See comment above that it is not best practices to combine across emission scenarios and decades – these should be reported separately.

Response: Thanking you for pointing out this! We have re-organized the figures and results by presenting the projections under different emission scenarios and decades separately. We have presented the RCP8.5 2050 projection as an example in the main figures (*i.e.*, Fig. 3c-d, Fig. 4 g-h and Fig. 5a-d), and placed other projections under different emission scenarios and decades separately in the supplementary figures (*i.e.*, Fig. S1, S4, S5, S6 and S7).

Q9. Fig. 2b – It is hard to see the different ranges and their overlap.

Response: We have added a separate panel to Fig. 2b showing the distribution ranges of the

two species and their overlap.

References:

- Anderson, R.P., Raza, A. 2010. The effect of the extent of the study region on GIS models of species geographic distributions and estimates of niche evolution: preliminary tests with montane rodents (genus *Nephelomys*) in Venezuela. *Journal of Biogeography*, 37, 1378–1393.
- Anderson, R.P., 2012. Harnessing the world's biodiversity data: promise and peril in ecological niche modeling of species distributions. *Annals of the New York Academy of Sciences* 1260, 66–80.
- Bay, R. A., Harrigan, R. J., Underwood, V. L., Gibbs, H. L., Smith, T. B., Rugg, K. (2018). Genomic signals of selection predict climate-driven population declines in a migratory bird. *Science*, 359, 83–86.
- Boria, R.A., Olson, L.E., Goodman, S. M., Anderson R. P. 2014. Spatial filtering to reduce sampling bias can improve the performance of ecological niche models. *Ecological Modelling*, 275, 73–77.
- Fitzpatrick, M. C., Keller, S. R. (2015). Ecological genomics meets community-level modelling of biodiversity: mapping the genomic landscape of current and future environmental adaptation. *Ecology Letters*, 18, 1–16.
- Pearson, R.G., Raxworthy, C., Nakamura, M., Peterson, A.T. (2007). Predicting species distributions from small numbers of occurrence records: a test case using cryptic geckos in Madagascar. *Journal of Biogeography*, 34, 102–117.
- Rugg, K., Bay, R. A., Anderson, E. C., Saracco, J. F., Harrigan, R. J., Whitfield, M., ..., Smith, T. B. (2018). Ecological genomics predicts climate vulnerability in an endangered southwestern songbird. *Ecology Letters*, 21, 1085–1096.

REVIEWERS' COMMENTS

Reviewer #1 (Remarks to the Author):

The authors have addressed all my comments adequately. I am happy to recommend the paper for publication.

Reviewer #2 (Remarks to the Author):

The authors have done a good job addressing the final concerns raised during the previous round of review and I appreciate their thoroughness and attention to detail. Overall, the study is clearer and substantially improved. I have only two minor points:

L53: As one of the original papers on the topic, Fitzpatrick & Keller (2015) needs to be cited here as well.

I raised the concern regarding the use of elevation in the GF models in the previous round of reviews. I would argue that the papers cited in the rebuttable that have also used elevation in GF models when predicting genomic offsets erred in doing so as well. That said, the authors do provide a convincing answer that models fit with or without elevation produce similar results, which is reassuring.

Reviewer #4 (Remarks to the Author):

Chen et al. have combined the ecological genomics and the niche modelling to evaluate the population specific responses of the two bird species in the Sino Himalayan Mountains. Their work highlighted the necessity of integrating genomic offset, niche suitability modelling, and landscape connectivity when assessing the climate change-driven vulnerability. Based on my research background, I mainly evaluated the bird genome assembly part of this paper. Overall, the authors have done a perfect job on this part. Only minor content clarifications are required.

Supplementary Note 1

This study used a 10X strategy to generate a de novo genome of *T. elliotii*, which is of comparable quality to published bird genomes. This assembly meets the quality requirement considering that the genome is subsequently used as a reference for SNP Calling.

Minor revision:

I would suggest the authors provide the following information:

- software used in de novo gene prediction
- software and the species used as the reference gene set in the homology-based method
- raw data, and assembly should also be released in public databases

Some inaccurate descriptions:

- “After gap-filling and removal of adaptor sequences, short reads were assembled ...”

Gap filling is not a step for processing the short reads. It is a finishing step in de novo genome assembly. Please reorganize this sentence.

- “under the pseudohap” -> “under the pseudohap style”
- “2.702Mb” -> “2.702 Mb”
- “Benchmarking Universal Single Copy Orthologs v2” -> Remove “v2”. The version number is written in the previous paragraph as “V. 3.0.2”.
- Please clarify the BUSCO score. Is it only based on the complete BUSCOs? Or does it also contain the fragmented BUSCOs?

Genomic data generation and processing

Minor revision:

- Please remove the redundant information in the line 436-440 when describing filtering criteria.
- Line 442: It should be qualified reads, not the raw data.

REVIEWERS' COMMENTS

Reviewer #1 (Remarks to the Author):

Q1. The authors have addressed all my comments adequately. I am happy to recommend the paper for publication.

Response: Thank you for the valuable comments to improve our work!

Reviewer #2 (Remarks to the Author):

Q1. The authors have done a good job addressing the final concerns raised during the previous round of review and I appreciate their thoroughness and attention to detail. Overall, the study is clearer and substantially improved. I have only two minor points:

L53: As one of the original papers on the topic, Fitzpatrick & Keller (2015) needs to be cited here as well.

Response: We have added Fitzpatrick & Keller 2015 (11) in Line 53.

Q2. I raised the concern regarding the use of elevation in the GF models in the previous round of reviews. I would argue that the papers cited in the rebuttable that have also used elevation in GF models when predicting genomic offsets erred in doing so as well. That said, the authors do provide a convincing answer that models fit with or without elevation produce similar results, which is reassuring.

Response: Thank you for raising this concern and we agree that the elevation shouldn't be included in the genomic offset prediction as it wouldn't change over time. We are delighted that you find our comparisons of models fit with and without elevations are convincing.

Reviewer #4 (Remarks to the Author):

Q1. Chen et al. have combined the ecological genomics and the niche modelling to evaluate the population specific responses of the two bird species in the Sino Himalayan Mountains. Their work highlighted the necessity of integrating genomic offset, niche suitability modelling, and landscape connectivity when assessing the climate change-driven vulnerability. Based on my research background, I mainly evaluated the bird genome assembly part of this paper. Overall, the authors have done a perfect job on this part. Only minor content clarifications are required.

Response: Thank you for your comments to improve our description of genome assembly and annotation.

Q2. Supplementary Note 1. This study used a 10X strategy to generate a de novo genome of T. elliotii, which is of comparable quality to published bird genomes. This assembly meets the quality requirement considering that the genome is subsequently used as a reference for SNP Calling.

Minor revision:

I would suggest the authors provide the following information:

- software used in de novo gene prediction; - software and the species used as the reference gene set in the homology-based method

Response: The description of genome assembly and annotation in Supplementary Note 1 in previous version has been moved to the Methods and Results in the main text. We have provided a detailed description of the methods related to de novo gene prediction, including software, and what species were used as reference gene set in lines 472-480, specifically as below.

“We applied the homolog-based approach to annotate the protein-coding genes by using the protein sequences of *Gallus gallus* and *Taeniopygia guttata*. The protein sequences of these reference genes were aligned to each genome using TABASTN (v2.2.26) (4) with an e-value cut off 1e-5, and multiple adjacent hits of the same query were connected by genBlastA v. 1.0.4 (5). Homologous blocks with length greater than 30% of the query protein length were retained. The connected hit region was later extended to include its 2 Kb flanking regions, on which gene structure was predicted by Genewise v.2.4.1 (6). We then used MUSCLE v3.8.31 (7) to align the annotated protein with the reference protein. Predicted proteins with length ≥ 30 amino acids and identity value $\geq 40\%$ were retained”.

Q3. - raw data, and assembly should also be released in public databases

Response: The genome assembly and sequencing data of the de novo sequenced individual of *T. elliotii* have been deposited to National Genomics Data center (<https://db.cngb.org/>) under BioProject accession CNP0003256 (<https://db.cngb.org/search/project/CNP0003256/>) and NCBI with

Bioproject	ID	PRJNA860040
------------	----	-------------

 (<https://www.ncbi.nlm.nih.gov/bioproject/?term=PRJNA860040>). We have provided this information in Data Availability (lines 762-766).

Q4. Some inaccurate descriptions:

- “After gap-filling and removal of adaptor sequences, short reads were assembled ...”

Gap filling is not a step for processing the short reads. It is a finishing step in de novo genome assembly. Please reorganize this sentence.

Response: We have removed “after gap-filling”.

Q5. - “under the pseudohap” -> “under the pseudohap style”

- “2.702Mb” -> “2.702 Mb”

Response: We have changed this.

Q6. - “Benchmarking Universal Single Copy Orthologs v2” -> Remove “v2”. The version number is written in the previous paragraph as “V. 3.0.2”.

Response: We have changed this.

Q7. - Please clarify the BUSCO score. Is it only based on the complete BUSCOs? Or does it also contain the fragmented BUSCOs?

Response: 92% only includes complete single-copy and duplicated BUSCOs. We previously hadn't presented result on fragmented BUSCOs, and we have now provided this part of information in the revised manuscript (lines 116-119), specifically as below.

"Using Benchmarking Universal Single-Copy Orthologs (BUSCO, aves_odb9) as the reference gene set, we estimated that the assembly contains 90% of the core genes as the complete single-copy BUSCOs, and 2% as the complete duplicated BUSCOs. Genome completeness based on fragmented BUSCOs is about 4.5%".

Q7. Genomic data generation and processing

Minor revision:

- Please remove the redundant information in the line 436-440 when describing filtering criteria.

Response: Thank you for noticing this. We have removed this redundant information.

Q8. - Line 442: It should be qualified reads, not the raw data.

Response: We have changed this.